# Virus infection of the CNS disrupts the immune-neural-synaptic axis via induction of pleiotropic gene regulation of host responses

Olga A Maximova[1][†]*, Daniel E Sturdevant[2], John C Kash[1], Kishore Kanakabandi[2], Yongli Xiao[1], Mahnaz Minai[3], Ian N Moore[3], Jeff Taubenberger[1], Craig Martens[2], Jeffrey I Cohen[1], Alexander G Pletnev[1]

[1]Laboratory of Infectious Diseases, National Institute of Allergy and Infectious Diseases, National Institutes of Health, Bethesda, United States; [2]Research Technologies Branch, Genomics Unit, National Institute of Allergy and Infectious Diseases, National Institutes of Health, Hamilton, United States; [3]Infectious Disease Pathogenesis Section, Comparative Medicine Branch, National Institute of Allergy and Infectious Diseases, National Institutes of Health, Bethesda, United States

**Abstract** Treatment for many viral infections of the central nervous system (CNS) remains only supportive. Here we address a remaining gap in our knowledge regarding how the CNS and immune systems interact during viral infection. By examining the regulation of the immune and nervous system processes in a nonhuman primate model of West Nile virus neurological disease, we show that virus infection disrupts the homeostasis of the immune-neural-synaptic axis via induction of pleiotropic genes with distinct functions in each component of the axis. This pleiotropic gene regulation suggests an unintended off-target negative impact of virus-induced host immune responses on the neurotransmission, which may be a common feature of various viral infections of the CNS.

*For correspondence:
maximovao@niaid.nih.gov

[†]Senior Author

**Competing interests:** The authors declare that no competing interests exist.

## Introduction

Many viruses can invade the central nervous system (CNS), infect the neurons, propagate by exploiting cellular machinery, hijack axonal transport, and cross the synapses to disseminate within neural networks (*Taylor and Enquist, 2015*). This is also true for the West Nile virus (WNV), which can infect the CNS, use bi-directional axonal transport, and spread trans-synaptically (*Maximova et al., 2016*). WNV infection has affected an estimated 7 million people in the continental United States since the first outbreak in 1999 (*Ronca et al., 2019*). In humans, WNV infection can result in WNV neurological disease (WNV-ND) (*Davis et al., 2006*) with a high in-hospital mortality rate (*Yeung et al., 2017*), motor and cognitive impairment (*Kleinschmidt-DeMasters and Beckham, 2015*; *Patel et al., 2015*), and increased risk of death during the convalescent phase (*Philpott et al., 2019*). WNV-ND can present as meningoencephalomyelitis with any combination of clinico-pathological features such as aseptic meningitis, encephalitis, and acute flaccid myelitis (*Sejvar, 2016*). The estimated ratio of the incidence of WNV-ND relative to the number of WNV-RNA-positive blood donors is ≤1% (*Betsem et al., 2017*), but the current burden of WNV-ND may be underestimated due to underutilization or inaccurate choice of diagnostic tests when viral encephalitis is clinically diagnosed (*Vanichanan et al., 2016*). The lessons from the past two decades of extensive research using mouse models have provided a better understanding of immune control of WNV infection (*Diamond and Gale, 2012*). However, current knowledge is yet to be translated into effective treatment or better

clinical management of WNV-ND, which remains only supportive. A major question of how virus replication in neurons and ensuing immune responses alter neural functions and result in neuropathological sequelae remains unanswered.

Postulated mechanisms of the neuropathogenic impact of WNV infection center around the loss of neurons due to either direct cytopathic effect of virus replication or damaging effects of the inflammatory environment developing in response to infection. Yet, in order to develop potential neuroprotective therapies, we need to better understand the mechanisms of neuronal dysfunction during acute infection, before neuronal loss has occurred, as well as the characteristics of neurological deficits after recovery. Loss of synapses in the hippocampus has been suggested as one of the potential mechanisms underlying cognitive impairment in patients recovering from WNV encephalitis (*Vasek et al., 2016*). The impact of WNV infection on neurophysiology is expected to be CNS-region-specific, which would result in the impairment of a function specific to an affected neural network. This necessitates a study of the pathogenic mechanisms of the impairment of motor functions in movement disorders, typically observed in WNV-ND (*Lenka et al., 2019*).

One unbiased approach to obtain a broad insight into complex pathophysiological changes occurring in the virus-infected CNS is to examine alterations in global gene expression. Although several studies have provided insights into changes in gene expression in the CNS after an experimental WNV infection, they mainly focused on the immune responses and apoptosis (*Lim et al., 2017*; *Kosch et al., 2018*; *Bourgeois et al., 2011*; *Clarke et al., 2014a*; *Venter et al., 2005*; *Green et al., 2016*; *Kumar et al., 2016*; *Clarke et al., 2014b*; *Vig et al., 2018*). Our understanding of the molecular mechanisms underlying changes in the regulation of the CNS during WNV-ND is incomplete. Here, we studied how the CNS is regulated when faced with a challenge to fight virus infection of neurons in a nonhuman primate (NHP) model, using combined analysis of the transcriptome, protein expression, and cell morphology.

## Results

Tissue samples for this study were selected from the CNS of NHPs inoculated intracerebrally with wild type WNV (NY99) (*Maximova et al., 2014*). The rationale for the selection of CNS structures for examination was as follows: (i) high viral burden at the site; (ii) severe histopathological changes at the site; and (iii) presence of neurological signs consistent with the function of the respective structure. Based on these criteria from the intracerebral NHP model (*Maximova et al., 2016*; *Maximova et al., 2014*), which are consistent with a body of data on the virology, histopathology, and clinical presentations of WNV-ND in humans (*Kleinschmidt-DeMasters and Beckham, 2015*; *Sejvar, 2016*; *Lenka et al., 2019*; *Omalu et al., 2003*; *Cushing et al., 2004*; *Guarner et al., 2004*; *Armah et al., 2007*; *Hart et al., 2014*), the cerebellum and spinal cord samples were selected for analysis. The samples were from the following predetermined time points after WNV infection: (i) 3 days post-infection (dpi; presymptomatic stage; n = 3); (ii) 7 dpi (early-symptomatic stage; moderate incoordination, tremor, and limb weakness; n = 3); and (iii) 9 dpi (advanced-symptomatic stage; severe incoordination, tremor, and limb weakness that required humane euthanasia; n = 3). The mock control cerebellum or spinal cord samples were from NHPs (n = 4) that were inoculated intracerebrally in an identical manner to the virus-inoculated animals, except that the inoculum contained only diluent and not virus (*Maximova et al., 2014*) (detailed procedure is described by us previously [*Maximova et al., 2008*]). These four mock-inoculated control animals were used for normalization of gene expression and one animal (euthanized at 10 dpi) was used as a normal control for immunohistochemistry that examined the advanced-symptomatic stage of WNV-ND (9 dpi).

### WNV infection of the CNS alters the homeostasis in transcriptional regulation of the immune and neural system domains

We examined gene expression changes in the cerebellar and spinal samples from the CNS of intracerebrally infected NHPs using Affymetrix microarray analysis and validated microarray results by qPCR. A broad examination of changes in global gene expression in response to WNV infection showed transcriptional changes that progressively increased during the neurological symptomatic stages in a CNS-region-specific manner (*Figure 1*, a and b; top 50 upregulated or downregulated genes are listed in c and d, respectively).

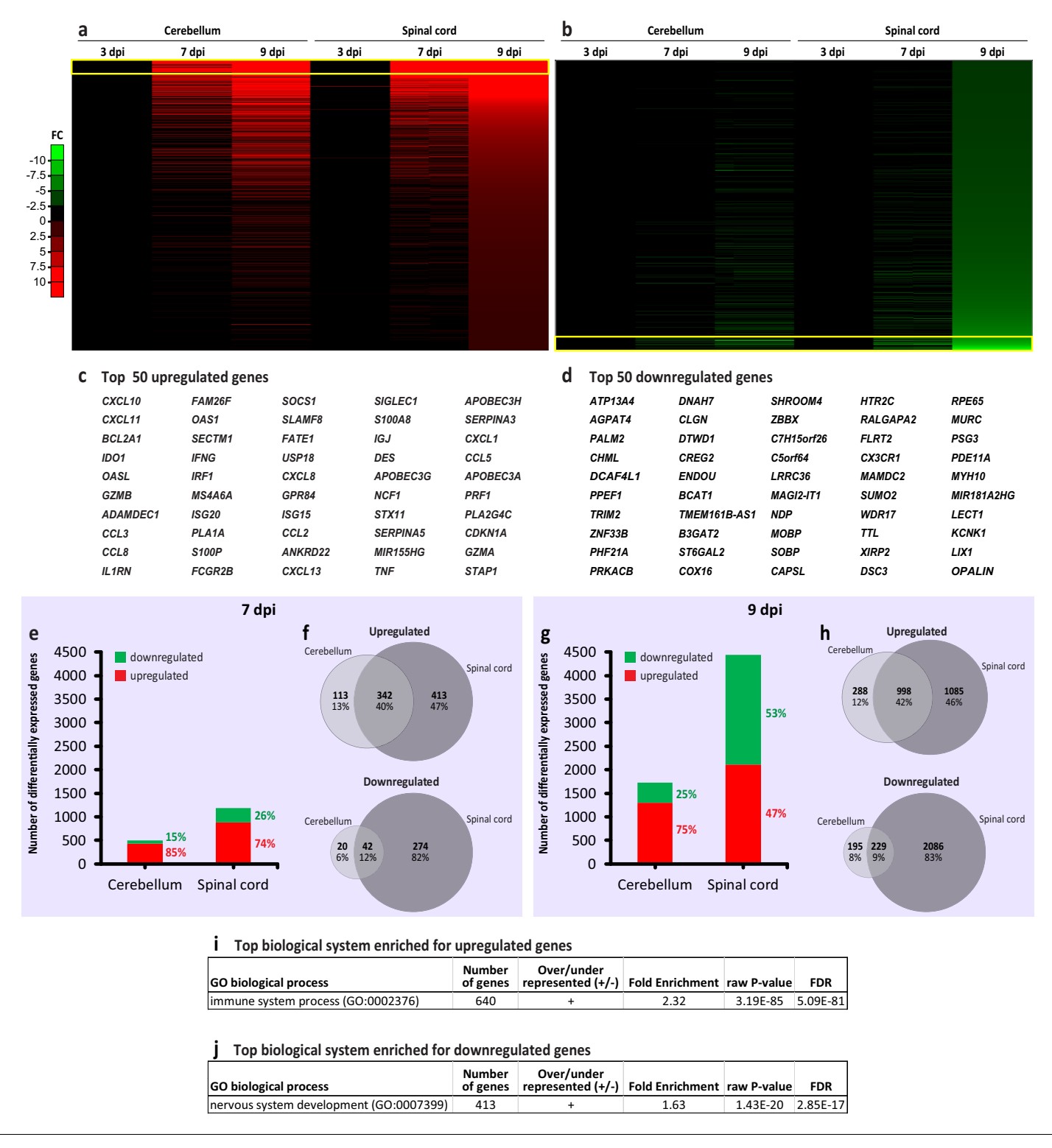

**Figure 1.** Transcriptional changes during progression of West Nile virus neurological disease (WNV-ND) converge on the immune and nervous system processes in a central nervous system (CNS)-region-specific manner. (**a and b**) Heatmaps for the upregulated (**a**) or downregulated (**b**) genes in the cerebellum and spinal cord at indicated days post-inoculation (dpi) with WNV (three animals per time point for each CNS region); based on the fold change (FC) over mock (one animal per time point, pooled). The arbitrary threshold for the visualization is −10 to 10 FC. (**c and d**) Listed are the top 50 upregulated or downregulated genes that correspond to the yellow boxed areas in **a** and **b**, respectively. (**e–h**) Ratios of the upregulated to downregulated genes (**e and g**) and Venn diagram comparison of the upregulated and downregulated genes (**f and h**) in the cerebellum and spinal

*Figure 1 continued on next page*

*Figure 1 continued*

cord during indicated stages of WNV-ND. (**i and j**) Identification of the top biological systems significantly enriched for the upregulated (**i**) or downregulated genes (**j**) by ORT. FDR, false discovery rate.

At the presymptomatic stage of WNV-ND (3 dpi), no genes expressed in the cerebellum were significantly differentially regulated compared to mock (false discovery rate, FDR > 0.05). In the spinal cord, however, six genes were significantly upregulated (FDR < 0.05; fold change ≥2 over mock) and none were significantly downregulated. The upregulated genes were: *MX1* (12.2-fold change), *IFIT3* (7.9-fold change), *IFI44L* (7.6-fold change), *PARP9* (4.0-fold change), *STAT1* (3.2-fold change), and *TRIM22* (2.1-fold change). Annotation of these genes to gene ontology (GO) terms revealed a functional dichotomy for some of these genes in regulating both the immune and nervous system. While all six genes were annotated to GO biological process (BP) terms such as defense response to virus (GO:0051607) and immune system process (GO:0002376), two of these genes (*MX1* and *STAT1*) were also annotated to the nervous system related terms such as the dendrite (GO:0030425), axon (GO:0030424), synapse (GO:0054202), and regulation of synapse structure or activity (GO: 0050803). These data suggest a dual immune-neural functionality of some early upregulated genes and offer a hint to a transcriptional activation of immune responses and concurrent modulation of neurotransmission, likely associated with sensing the spread of WNV within the CNS, which can occur trans-synaptically (*Maximova et al., 2016*). Since the small number of the differentially regulated genes precluded the use of statistical analyses for this presymptomatic stage, all functional genomic analyses hereafter will be focused on the early-symptomatic (7 dpi) and/or advanced-symptomatic (9 dpi) stages of WNV-ND.

At the early-symptomatic stage of WNV-ND (7 dpi), there were many more upregulated than downregulated genes in both the cerebellum and spinal cord, with much stronger transcriptional changes in the spinal cord (*Figure 1e*). At the advanced-symptomatic stage of WNV-ND (9 dpi), compared to early-symptomatic stage, the number of upregulated and downregulated genes increased in both the cerebellum and spinal cord (*Figure 1f*), with about twice as many differentially expressed genes (DEGs) in the spinal cord, compared to the cerebellum. Notably, the spinal cord also consistently displayed a higher percentage of the downregulated genes, which doubled at the advanced symptomatic stage compared to the early-symptomatic stage (*Figure 1e and f*). Similarity between the genes that were differentially expressed in the WNV-infected cerebellum and spinal cord was higher for the upregulated genes (40–42%), compared to overlaps in the downregulated genes (9–12%) (*Figure 1f and h*).

The overrepresentation test (ORT; BP complete; FDR < 0.05), using the lists of genes that were upregulated or downregulated in the spinal cord at the advanced-symptomatic stage, identified the immune and nervous systems as the top biological systems significantly enriched for the upregulated (*Figure 1g*) and downregulated genes (*Figure 1h*), respectively.

Since the top largest enriched GO BP term for downregulated genes was 'nervous system development', we further dissected transcriptional regulation of the developmental and repair processes in the CNS that may be altered by WNV infection using the gProfiler (*Raudvere et al., 2019*). Analysis of the upregulated genes in the cerebellum and spinal cord revealed upregulation of the developmental processes with major 'spikes' (i.e., most significant enrichment) related to the immune system development, response to wounding, and wound healing (*Figure 2a and c*). Both the cerebellum and spinal cord also displayed upregulation of the gliogenesis/astrocyte development and extracellular matrix/tissue remodeling. In contrast, developmental and/or repair processes related to neurons, their cellular components, and respective anatomical structure/network development were significantly enriched for the downregulated genes, with major 'spikes' related to neuron/neuron projection development and axon development/guidance (*Figure 2b and d*). This suggested that there was a downregulation rather than activation of the developmental processes related to neurons. However, given the detected spikes in regulation of axon development/guidance, we asked whether the molecular environment for axonal regeneration was inhibitory or permissive based on the up- or downregulation of expression of established axon growth inhibitory or permissive molecules (*Anderson et al., 2016*). Of 59 axon-growth-modulating molecules analyzed, we found changes in gene expression for 6 inhibitory and 16 permissive molecules (*Figure 2e*). Taking into account the fold changes for these 22 axon growth genes, when (i) downregulation of permissive or

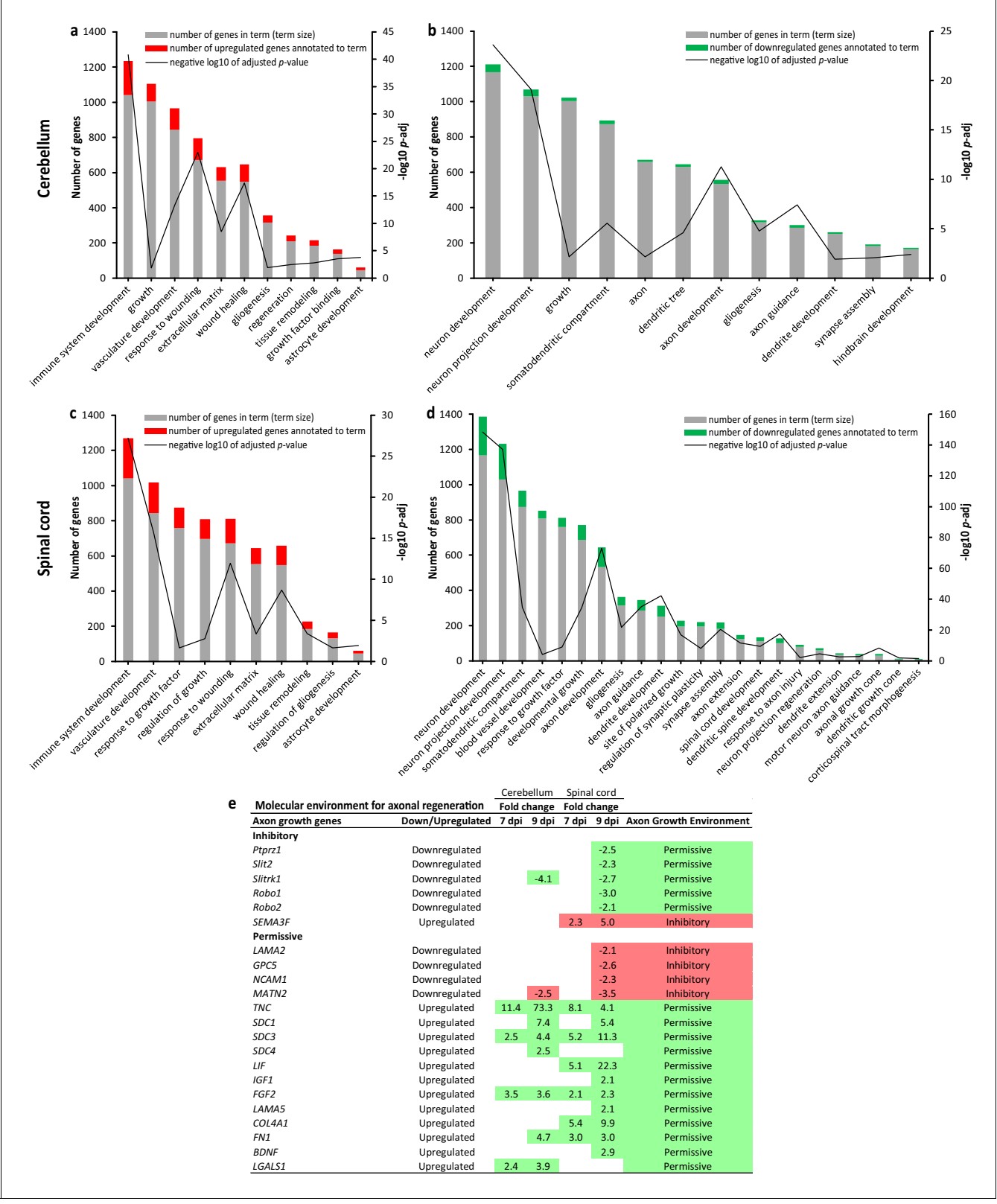

**Figure 2.** West Nile virus (WNV) infection alters transcriptional regulation of the developmental and repair processes in the central nervous system (CNS). (**a–d**) Enrichment of the developmental process and/or repair in WNV-infected cerebellum (**a and b**) and spinal cord (**c and d**) based on the ratios between GO term size (number of genes in term) and intersection size (number of differentially expressed genes annotated to term) at the

*Figure 2 continued on next page*

*Figure 2 continued*

advanced-symptomatic stage of WNV neurological disease (WNV-ND; 9 dpi). (**e**) Dissection of the molecular environment for axonal regeneration based on the differential transcriptional regulation of the molecules with established inhibitory or permissive roles in the regulation of axon growth.

upregulation of inhibitory molecules would indicate the inhibitory molecular environment for axonal regeneration, while (ii) downregulation of inhibitory or upregulation of permissive molecules would indicate the permissive environment, we found a progressive trend which was skewed to the axon growth permissive environment.

Next, we used the Reactome Knowledgebase (*Fabregat et al., 2018*) to identify the biological pathways enriched in immune and neural system domains. The Reactome Knowledgebase currently has 27 annotated biological domains, including the immune and neuronal systems, that are visualized as the pathway hierarchy bursts (*Figure 3a and b*) and can be overlaid by the identified significantly affected pathways to display their coverage.

As expected based on the identification of the immune system as the top biological system significantly enriched for the upregulated genes (*Figure 1g*), Reactome pathways associated with this system showed significant enrichment only for upregulated genes (*Figure 3c, d, g, and h*), but not for downregulated genes (at both early-symptomatic and advanced-symptomatic stages of WNV-ND, and in both the cerebellum and spinal cord). Major immune system subdomains such as the innate immune system (including the toll-like receptor cascades), cytokine signaling (including interferon signaling and signaling by interleukins), and the adaptive immune system (including class I MHC mediated antigen processing and presentation, and TCR/BCR receptor signaling) were identified as significantly enriched in both the cerebellum and spinal cord of NHPs at the symptomatic stages of WNV-ND (*Figure 3c, d, g, and h*). The only difference between the cerebellum and spinal cord in terms of activated immune pathways was that MAP kinase activation reached statistical significance only in the spinal cord at the advanced-symptomatic stage of WNV-ND (compare *Figure 3d and h*; pathway node #12).

As expected based on the identification of the neuronal system as the top biological system significantly enriched for the downregulated genes (*Figure 1h*), Reactome pathways associated with this system showed significant enrichment only for downregulated genes (*Figure 3e, f, i, and j*), but not for upregulated genes (at both early-symptomatic and advanced-symptomatic stages of WNV-ND, and in both the cerebellum and spinal cord). The neuronal system domain was most affected at the advanced-symptomatic stage in both the cerebellum and spinal cord (compare *Figure 3e, f, i, and j*), coinciding with the progression to severe neurological signs at this stage (*Maximova et al., 2014*). We noted that the Reactome's neuronal system domain is limited only to pathways associated with the neurotransmission and does not cover the complexity of the nervous system. Nonetheless, it enabled identification of transmission across chemical synapses as a common significantly affected pathway in the cerebellum and spinal cord (*Figure 3f, and j*; pathway node #2). The only difference between the cerebellum and spinal cord in terms of affected neuronal pathways was that the protein–protein interactions at synapses reached the statistical significance only in the spinal cord at the advanced-symptomatic stage of WNV-ND (compare *Figure 3f and j*; pathway node #8).

Together, these results show that WNV infection induces progressive and CNS-structure-dependent gene expression changes that manifest in transcriptional upregulation of biological pathways ascribed to the immune system gene ontologies and downregulation of functions related to the neuronal system biological domain. Therefore, a larger overlap in the upregulated genes compared to downregulated genes between the cerebellum and spinal cord may be explained by a higher similarity in the immune responses, but a site-specific impairment of distinct neural functions carried by these two CNS structures. In addition, we show that WNV infection alters transcriptional regulation of developmental and repair processes in the CNS, which are associated with upregulation of the immune system development/wound healing and downregulation of the neuronal/axonal regenerative processes. However, further dissection of the molecular environment associated with axonal regeneration also indicated a trend that was more skewed to the axon growth permissive rather than inhibitory environment, suggesting initiation of the neuronal network repair programs.

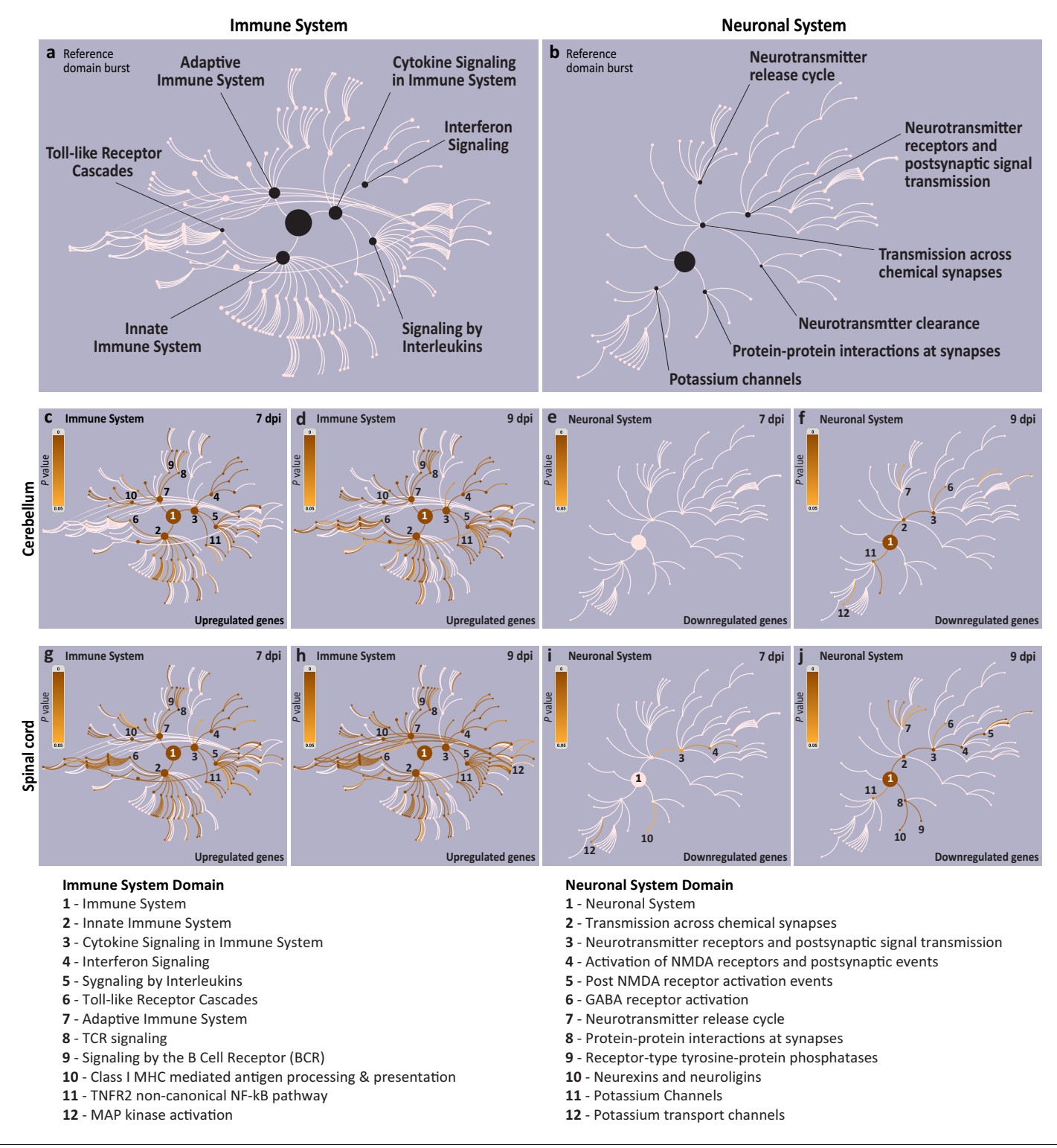

**Immune System Domain**
1 - Immune System
2 - Innate Immune System
3 - Cytokine Signaling in Immune System
4 - Interferon Signaling
5 - Sygnaling by Interleukins
6 - Toll-like Receptor Cascades
7 - Adaptive Immune System
8 - TCR signaling
9 - Signaling by the B Cell Receptor (BCR)
10 - Class I MHC mediated antigen processing & presentation
11 - TNFR2 non-canonical NF-kB pathway
12 - MAP kinase activation

**Neuronal System Domain**
1 - Neuronal System
2 - Transmission across chemical synapses
3 - Neurotransmitter receptors and postsynaptic signal transmission
4 - Activation of NMDA receptors and postsynaptic events
5 - Post NMDA receptor activation events
6 - GABA receptor activation
7 - Neurotransmitter release cycle
8 - Protein-protein interactions at synapses
9 - Receptor-type tyrosine-protein phosphatases
10 - Neurexins and neuroligins
11 - Potassium Channels
12 - Potassium transport channels

**Figure 3.** Visualization of the affected immune and neuronal system pathways during symptomatic stages of West Nile virus neurological disease (WNV-ND). (a and b) Reference Reactome biological domain bursts for the immune system (a) and neuronal system (b). Each reference burst visualizes the pathways specific to a respective biological system. The largest central node of each burst corresponds to the uppermost level of the domain hierarchy and successive concentrically positioned nodes and arcs represent more specific pathway levels. The major pathway nodes are indicated for each system. (c–j) Diagrams illustrate the coverage of reference bursts for the immune system (c), (d), (g), and (h) or neuronal system (e), (f), (i), and (j) by pathways affected during the early-symptomatic (7 dpi) and advanced-symptomatic (9 dpi) stages of WNV-ND in the cerebellum (c–f) or spinal cord (g–

*Figure 3 continued on next page*

*Figure 3 continued*

j). Each diagram (**c–j**) displays the enriched pathway coverage by overlaying the reference domain bursts (light pink) with an orange gradient color based on the p-values derived from an overrepresentation test (reference gradient is provided in each diagram based on the range of p-values from 0 to 0.05; darker colors indicate smaller p-values). Select nodes/pathways are enumerated and listed in the legends at the bottom of the figure.

## Altered CNS homeostasis in WNV-ND is defined by the large and subtle coordinated transcriptional shifts regulating the immune and nervous system, respectively

To further analyze the magnitude of transcriptional dysregulation within the CNS caused by WNV infection, we employed a statistical functional enrichment method that takes into account the fold changes associated with DEGs (*Mi et al., 2019*). This method has an advantage of detecting the coordinated changes in expression of groups of genes that are shifted from the overall distribution. Such coordinated transcriptional shifts can be subtle and elude other functional annotation methods (*Clark et al., 2003*; *Mootha et al., 2003*). We reasoned that this method could provide a better understanding of the magnitude of transcriptional dysregulation of the nervous system that appeared relatively subtle at the level of pathway analysis (*Figure 3*). Since the pathway level analysis showed that the immune and neuronal domains were affected the most at the advanced-symptomatic stage of WNV-ND, we analyzed genes that were significantly differentially expressed in WNV-infected cerebellum or spinal cord at this time point, together with their expression values (FC over mock). The results of this test, as expected, revealed coordinated large and subtle transcriptional shifts regulating the immune and nervous system, respectively. The large coordinated shifts (i. e., shifts to the greater fold change values from the overall distribution) were associated with the host defense response to viral infection and innate/adaptive immune responses (*Figure 4a–d*). In contrast, the subtle coordinated shifts (i.e., shifts to the smaller fold change values from the overall distribution) were associated with the cellular compartments of neurons (i.e., neuronal cell body, dendrite, axon, and glutamatergic and GABA-ergic synapses) and functions related to neurons (i.e., chemical synaptic transmission, neurotransmitter secretion, action potential, ensheathment of neurons, microtubule cytoskeleton organization, and synapse organization) (*Figure 4e–h*). The complete results are provided in *Supplementary file 1* ('Coordinated transcriptional shifts').

These results demonstrate that WNV infection induces coordinated shifts in the transcriptional regulation of the CNS homeostasis and that these shifts differ in magnitude and are large when associated with the immune system and subtle when related to the nervous system.

## Cell death processes in WNV-ND are regulated bi-directionally and magnitude of neuron cell death regulation does not exceed that of lymphocytes

Since the subtle coordinated transcriptional shifts in WNV-ND were associated with neurons and their functions (*Figure 4e–h*), but large shifts also included the regulation of cell death (*Figure 4b and d*), we next used the gProfiler (*Raudvere et al., 2019*) to simultaneously dissect the GO terms of BP and KEGG pathways (*Kanehisa et al., 2019*) to gain insight into the magnitude and direction of the regulation of cell death processes at the advanced-symptomatic stage of WNV-ND, with a particular focus on neurons.

Analysis of the genes that were downregulated in the cerebellum or spinal cord did not reveal any significantly enriched terms related to cell death, apoptosis, necroptosis, or necrosis. In contrast, analysis of the genes that were upregulated in the cerebellum or spinal cord showed significant (FDR < 0.05) enrichment of terms associated with regulation of cell death. Further dissection of the transcriptional regulation of cell death processes in WNV-ND revealed that: (i) cell death processes in WNV-ND were regulated bi-directionally (i.e., enriched GO BP terms included both negative and positive regulation) (*Figure 5a*); (ii) negative regulation of cell death appeared to outweigh the positive regulation based on the larger number of annotated genes in both the cerebellum and spinal cord (*Figure 5a*), although the FDR-adjusted p-values for the negative and positive regulation were both highly significant and comparable (in the range of 12.2–18.7 negative log10 of adjusted p-value); (iii) regulation of the following types of the cell death was significantly enriched: necroptosis, apoptosis, and necrosis (*Figure 5a*); (iv) regulation of the cell death processes was significantly

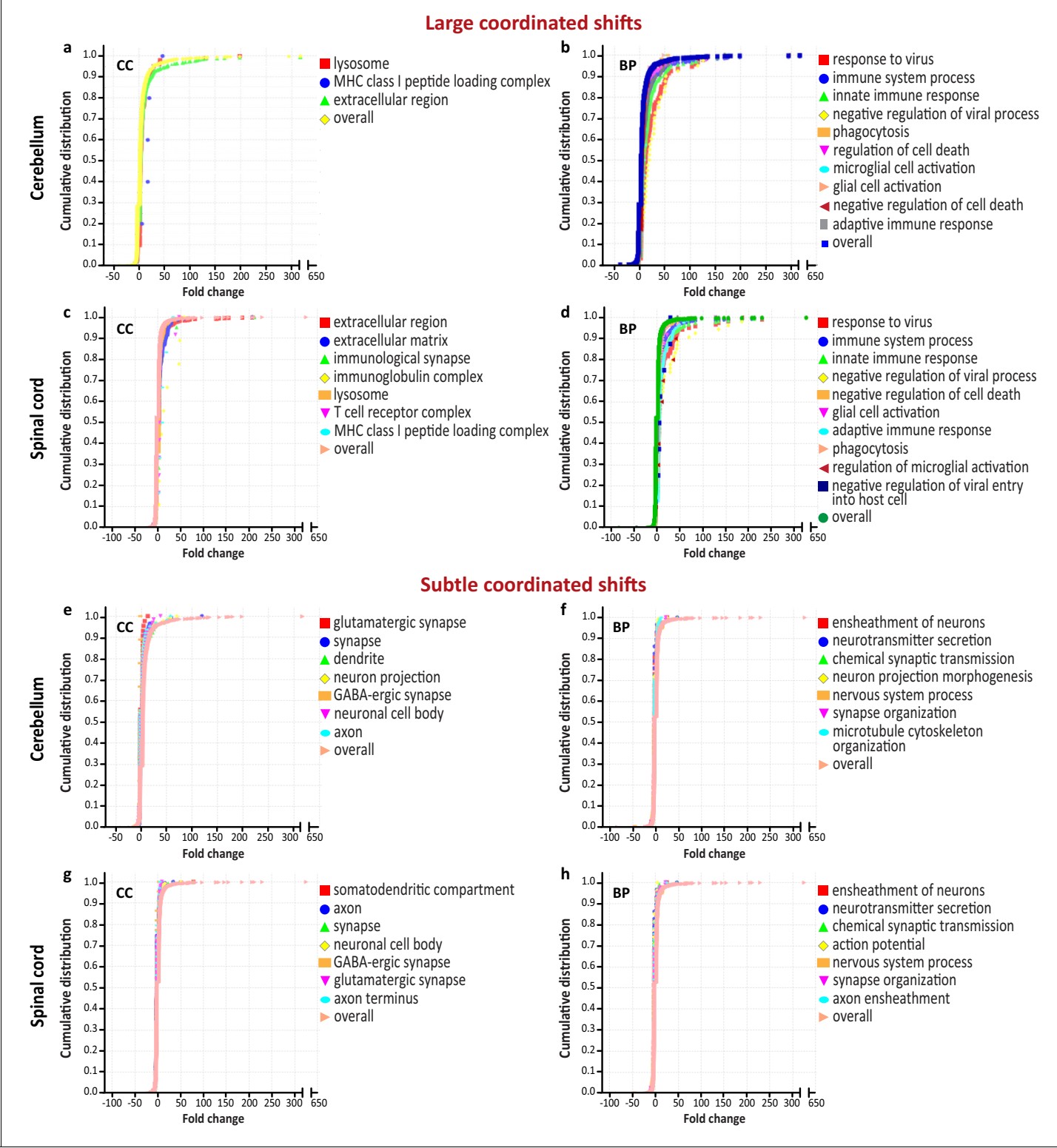

**Figure 4.** Coordinated transcriptional shifts during the advanced-symptomatic stage of West Nile virus neurological disease (WNV-ND). (a–h) Graphs show the cumulative distribution of fold change values for groups of differentially expressed genes and select large coordinated shifts from the overall distribution (predominantly under the curve) (a–d) or subtle coordinated shifts from the overall distribution (predominantly above the curve) (e–h) in the cerebellum (a), (b), (e), and (f) and spinal cord (c), (d), (g), and (h). Significantly enriched GO terms (FDR < 0.05) for the plotted groups of genes are shown in the corresponding legends. BP, biological process; CC, cellular component; GO, gene ontology.

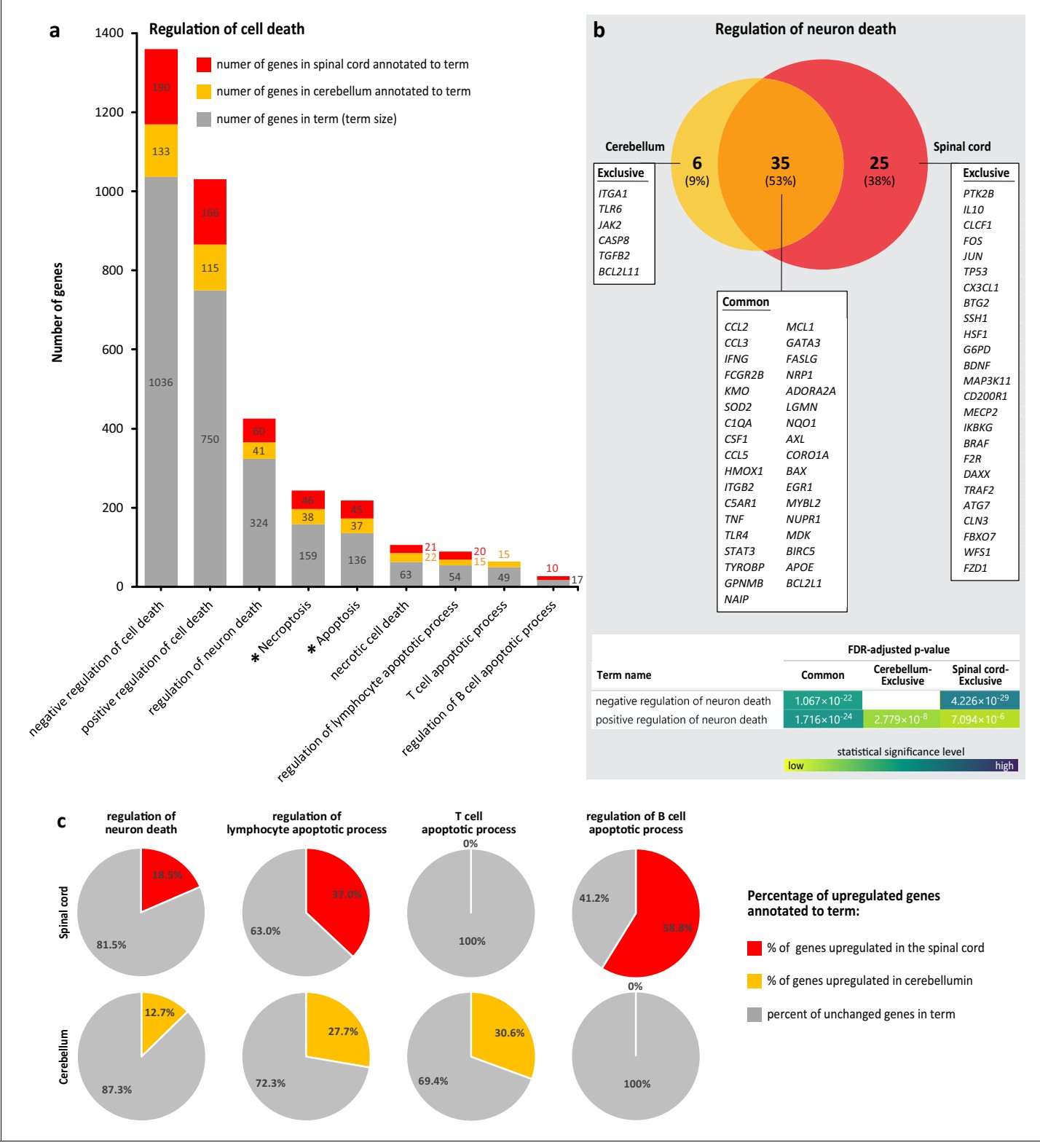

**Figure 5.** Dissection of transcriptional regulation of cell death processes during advanced-symptomatic West Nile virus neurological disease (WNV-ND). (a) Comparative analysis of the number of upregulated genes annotated to significantly enriched (FDR < 0.05) terms associated with cell death processes (BP GO terms) and pathways (KEGG pathways; asterisks) in the cerebellum and spinal cord (numbers of upregulated genes annotated to each term and term sizes are indicated). (b) Venn diagram comparison of the upregulated genes annotated to the term 'regulation of neuron death' in the cerebellum and spinal cord with functional characterization of each indicated gene set (gene symbols and statistical details for enriched GO BP

*Figure 5 continued on next page*

*Figure 5 continued*
terms are provided for each component of Venn diagram; blank: not significant). (c) Percentages of genes in indicated cell-specific terms of interest that were upregulated in WNV-ND relative to the term size (calculated from data in a).

enriched in two types of cells: neurons and lymphocytes (*Figure 5a*); (v) 53% of genes annotated to the 'regulation of neuron death' were commonly upregulated in the cerebellum and spinal cord, with only 9% of annotated genes exclusively upregulated in the cerebellum and 38% exclusively upregulated in the spinal cord (*Figure 5b*); (vi) functional characterization of the common gene set annotated to regulation of neuron death showed enrichment for both negative and positive regulation with a similar level of statistical significance, but genes that were regulated exclusively in the cerebellum contributed more to positive regulation while genes exclusively regulated in the spinal cord more significantly contributed to the negative regulation (*Figure 5b*); (vii) the percent of genes that became upregulated relative to the total number of genes known to be involved in regulation of the neuron death (i.e., term size) was about twice as a small than that for genes involved in regulation of the apoptotic process in lymphocytes, in both the cerebellum and spinal cord (*Figure 5c*); and (viii) regulation of the lymphocyte apoptotic processes was a region-specific (in the cerebellum regulation of apoptosis was increased in T cells, while in the spinal cord regulation of apoptosis was increased in B cells) (*Figure 5c*).

Taken together, these results demonstrate that activation of transcriptional regulation of the cell death in WNV-ND is (i) bi-directional (i.e., concurrently positive and negative), (ii) region-specific (i.e., skewed to a positive regulation in the cerebellum but negative regulation in the spinal cord), and (iii) cell-type-specific (i.e., major cell types with increased regulation of the cell death processes were neurons and lymphocytes). Importantly, the magnitude of cell death regulation in neurons was about twofold less than that in lymphocytes, suggesting that neurons are not a major cell type with increased transcriptional regulation of cell death processes. Instead, it appears that the major cell type with increased regulation of cell death processes was represented by the lymphocytes (T and B cells that are expected to infiltrate the CNS parenchyma as a part of the adaptive immune response to viral infection [*Maximova and Pletnev, 2018*]). Strikingly, regulation of the apoptotic processes in lymphocytes was region-specific, with increased regulation of T-cell apoptosis in the cerebellum but not in the spinal cord and vice versa – increased regulation of B-cell apoptosis in the spinal cord but not in the cerebellum, suggesting differential lymphocytic responses to WNV infection in these two CNS regions.

### WNV-induced transcriptional changes correspond to activation of non-neuronal multicellular responses to infection and disruption of the structural integrity and function of infected neurons

#### Large coordinated transcriptional shifts correspond to WNV-induced activation of non-neuronal multicellular responses to infection by CNS-intrinsic and CNS-extrinsic cells

Based on the identified large coordinated shifts in transcriptional regulation of the CNS homeostasis in WMV-ND (*Figure 4a–d*; *Supplementary file 1* ['Coordinated transcriptional shifts']), we selected three major BPs and examined the infected tissues by immunohistochemistry, using appropriate protein markers. These processes were: (i) reactive astrocytosis; (ii) microglia/macrophage activation and migration; and (iii) lymphocytic infiltration and migration (*Table 1*).

Immunoreactivity for glial fibrillary acidic protein (GFAP) was increased in the cerebellum and spinal cord in advanced WNV-ND and showed hypertrophy of somata and processes of astrocytes and their neuron-centripetal migration and perineuronal topology, all consistent with reactive astrocytosis (response of CNS-intrinsic glial cells to damage and disease [*Burda and Sofroniew, 2014*], including flavivirus infections [*Maximova and Pletnev, 2018*]; *Figure 6a*). These changes in astrocyte morphology and topology corresponded to the significantly enriched GO terms related to glial cell activation and regulation of cell migration (*Table 1*).

Strong immunoreactivity for the lysosomal-associated membrane protein CD68 (undetectable under normal physiological conditions) was observed in the cerebellum and spinal cord in advanced WNV-ND, consistent with microglial/macrophage activation. Activated/reactive microglial cells

**Table 1.** Dissection of West Nile virus (WNV)-induced coordinated transcriptional shifts at the protein marker and cell morphology/topology/function levels.

| Biological process | Enriched GO terms | Protein marker IR* | Cellular morphology/topology/function | Representative images |
|---|---|---|---|---|
| Reactive astrocytosis | glial cell activation (GO:0061900); regulation of cell migration (GO:0030334) | GFAP ↑ | *Astrocytes*: hypertrophy of somata and processes; neuron-centripetal migration; perineuronal topology | *Figure 6a* |
| Microglia/macrophage activation and migration | innate immune response (GO:0045087); glial cell activation (GO:0061900); regulation of cell migration (GO:0030334); regulation of microglial cell activation (GO:1903978); macrophage activation (GO:0042116); regulation of macrophage chemotaxis (GO:0010758); phagocytosis (GO:0006909); lysosome (GO:0005764) | CD68 ↑ | *Microglial cells*: hypertrophy of somata and processes; neuron-centripetal migration; perineuronal topology; phagocytic activity | *Figure 6b* |
| Lymphocytic activation, migration, and infiltration | adaptive immune response (GO:0002250); lymphocyte activation (GO:0046649); lymphocyte migration (GO:0072676); B cell mediated immunity (GO:0019724); regulation of T cell mediated immunity (GO:0002709); regulation of CD4-positive, alpha-beta T cell activation (GO:2000514); regulation of CD8-positive, alpha-beta T cell activation (GO:2001185) | CD20 ↑ | *B cells*: leptomeningeal infiltration in the cerebellar cortex; perivascular infiltration if the spinal cord gray matter; no parenchymal migration | *Figure 6c* |
| | | CD4 ↑ | *T helper cells*: leptomeningeal infiltration; perivascular infiltration; minimal parenchymal migration | *Figure 6c* |
| | | CD8 ↑ | *Cytotoxic T lymphocytes*: leptomeningeal infiltration; perivascular infiltration; parenchymal infiltration; neuron-centripetal migration; perineuronal topology | *Figure 6c* |
| Virus infection of specific neuronal cell types | negative regulation of viral process (GO:0048525); regulation of viral entry into host cell (GO:0046596) | WNV-Ag positivity | *Purkinje cells (PCs; CALB-positive)*: presence of viral antigens in neuronal perikarya and processes<br>*Spinal motor neurons (SMNs; ChAT-positive)*: presence of viral antigens in neuronal perikarya and processes | *Figure 7a and b* *Figure 7c and d* |
| Disruption of structural integrity and function of infected neurons | neuronal cell body (GO:0043025); somatodendritic compartment (GO:0036477); dendrite (GO:0030425); regulation of cytosolic calcium ion concentration (GO:0051480); cellular calcium ion homeostasis (GO:0006874); calcium ion-regulated exocytosis of neurotransmitter (GO:0048791); neurotransmitter secretion (GO:0007269); synapse (GO:0045202); neuron to neuron synapse (GO:0098984); presynapse (GO:0098793); asymmetric synapse (GO:0032279); distal axon (GO:0150034); glutamatergic synapse (GO:0098978); GABA-ergic synapse (GO:0098982); synapse organization (GO:0050808); synaptic signaling (GO:0099536); chemical synaptic transmission (GO:0007268) | CALB ↓ | *Purkinje cells*: decrease of CALB-IR in somatodendritic compartments of WNV-infected PCs; disturbance of cellular calcium homeostasis in infected PCs | *Figure 7b* |
| | | ChAT ↓ | *Spinal motor neurons*: decrease of ChAT-IR in SMN cytoplasm and synapses innervating SMNs; disruption of neurotransmitter secretion in SMNs and their afferent innervation | *Figure 7d* |
| | | SYP ↓ | Decrease in IR for structural constituent of synaptic vesicles (SYP-IR) in the cerebellar cortex and spinal cord. Disruption of structural integrity of presynaptic compartments.<br>*Cerebellar cortex*: putatively affected synapses innervating PCs: CF-PC and PF-PC (asymmetric-glutamatergic) in ML; BC-PC (GABA-ergic) in ML and PCL; and SC-PC (GABA-ergic) in ML<br>*Spinal gray matter*: putatively affected synapses innervating SMNs: cholinergic C- boutons (ChAT-IR); asymmetric-glutamatergic synapses from descending tracts; and inhibitory synapses from local inhibitory neuron networks | *Figure 8a and c* *Figure 8a* *Figure 7d* and *Figure 8c* |
| | neuronal cell body (GO:0043025); somatodendritic compartment (GO:0036477); dendrite (GO:0030425); postsynapse (GO:0098794); microtubule cytoskeleton organization (GO:0000226) | MAP2 ↓ | *Cerebellar cortex and spinal gray matter*: loss of postsynaptic cellular targets for asymmetric (axodendritic) and symmetric (axosomatic) synapses; disruption of integrity of microtubule cytoskeleton | *Figure 8b and d* |

*IR, immunoreactivity. ↑ - increased compared to mock. ↓ - decreased compared to mock. GFAP, glial fibrillary acidic protein. WNV-Ag, WNV antigens. CALB, calbindin D28k. ChAT, choline acetyltransferase. SYP, synaptophysin. MAP2, microtubule associated protein 2. ML. molecular layer. PCL, Purkinje cell (PC) layer. CF-PC, climbing fiber to PC synapses. PF-PC, parallel fiber to PC synapses. BC-PC basket cell to PC synapses. SC-PC, stellate cell to PC synapses.

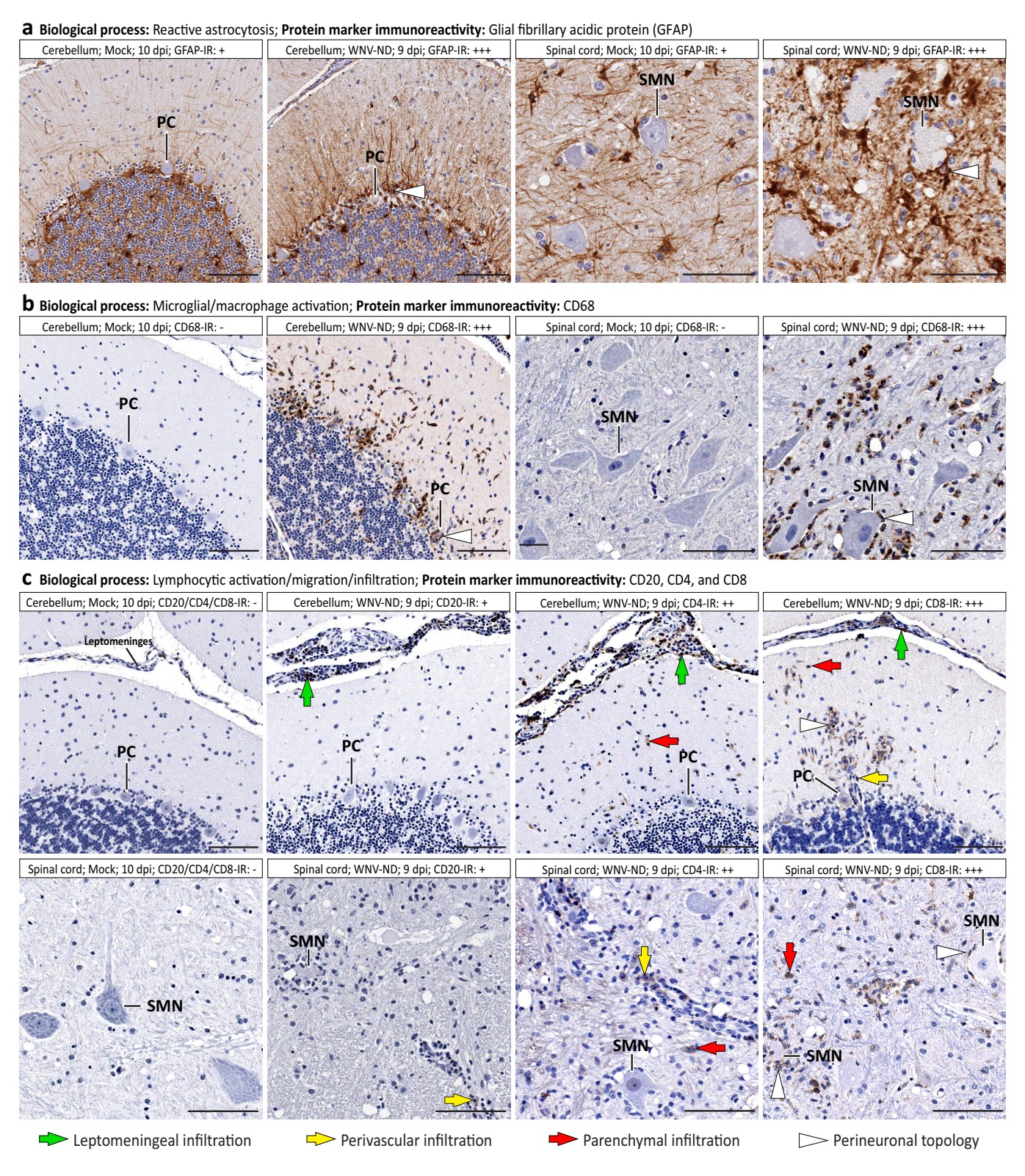

**Figure 6.** Immune cell morphology and topology during advanced-symptomatic West Nile virus neurological disease (WNV-ND). Representative images illustrate major cellular immune responses (indicated in **a–c**) by displaying immunoreactivity (IR) for relevant protein markers (brown) in WNV-infected cerebellum and spinal cord versus mock. Labeling keys are provided at the bottom of the figure. Semi-quantitative assessment of the IR is as follows: -, negative; +, minimal; ++; moderate; +++, strong. PC, Purkinje cell. SMN, spinal motor neuron. Scale bars: 100 μm.

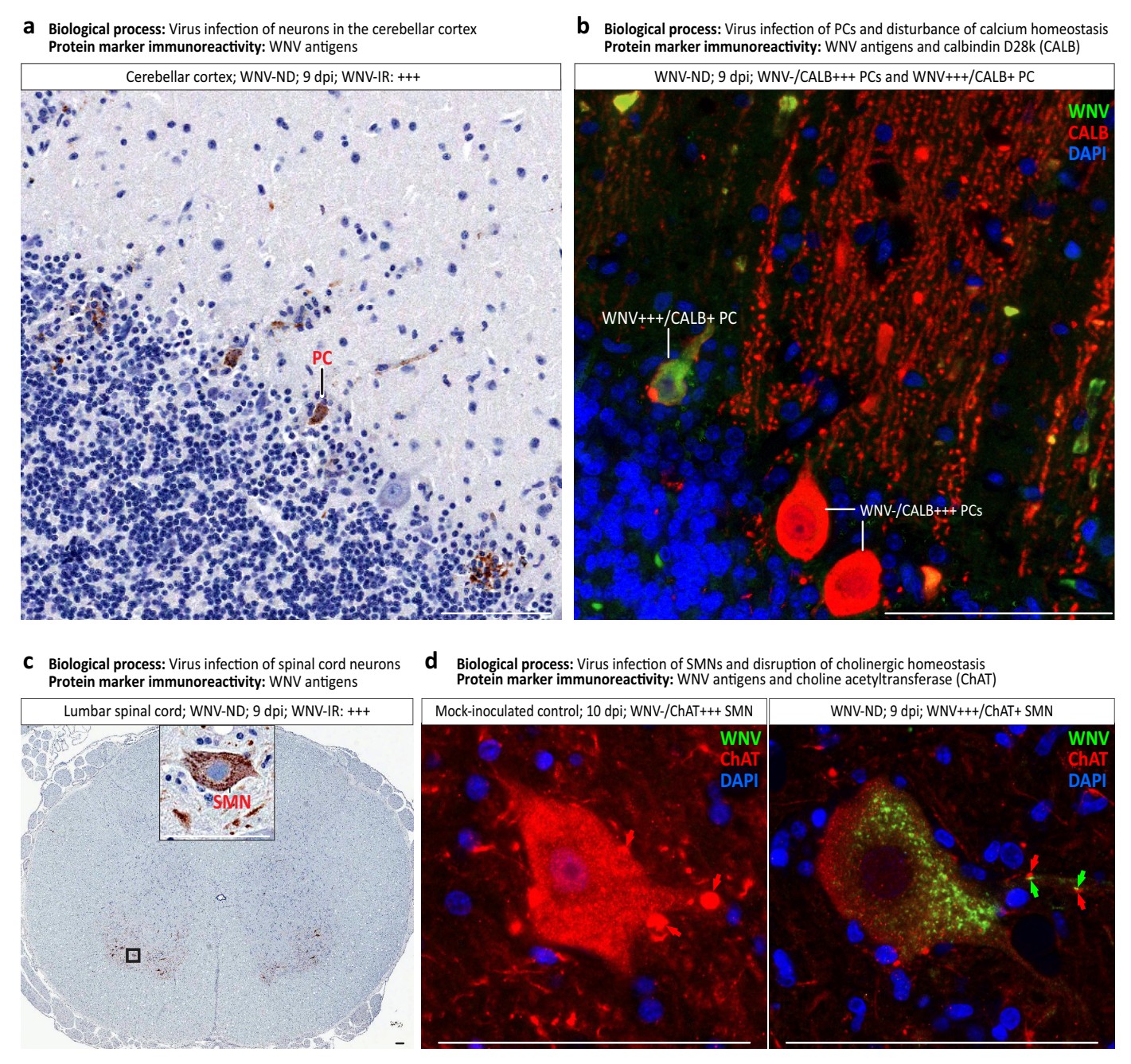

**Figure 7.** Virus-infected neuronal cell types and loss of neuronal cell-specific protein markers in West Nile virus neurological disease (WNV-ND). (a–d) Representative images illustrate identification of the types of neurons infected with WNV in the cerebellar cortex (a and b) and ventral horns of the gray matter in the spinal cord (c and d). Viral infection of specific neuronal types as a major biological process, and immunoreactivity (IR) for each protein marker ((a) and (c): brown; (b) and (d): colors are indicated in top-right corners) are provided for each panel. Red arrows in (d) indicate the ChAT-positive cholinergic presynaptic C-boutons innervating the somata and proximal dendrites of SMNs. Green arrows in (d) point to focal accumulations of WNV+++ granules in the proximal dendrites of WNV-infected SMN. Note that WNV+++ granules are immediately adjacent to few remaining ChAT+ C-boutons. Semi-quantitative assessment of the IR is as follows: -, negative; +, minimal; ++; moderate; +++, strong. PC, Purkinje cell. SMN, spinal motor neuron. Scale bars: 100 µm.

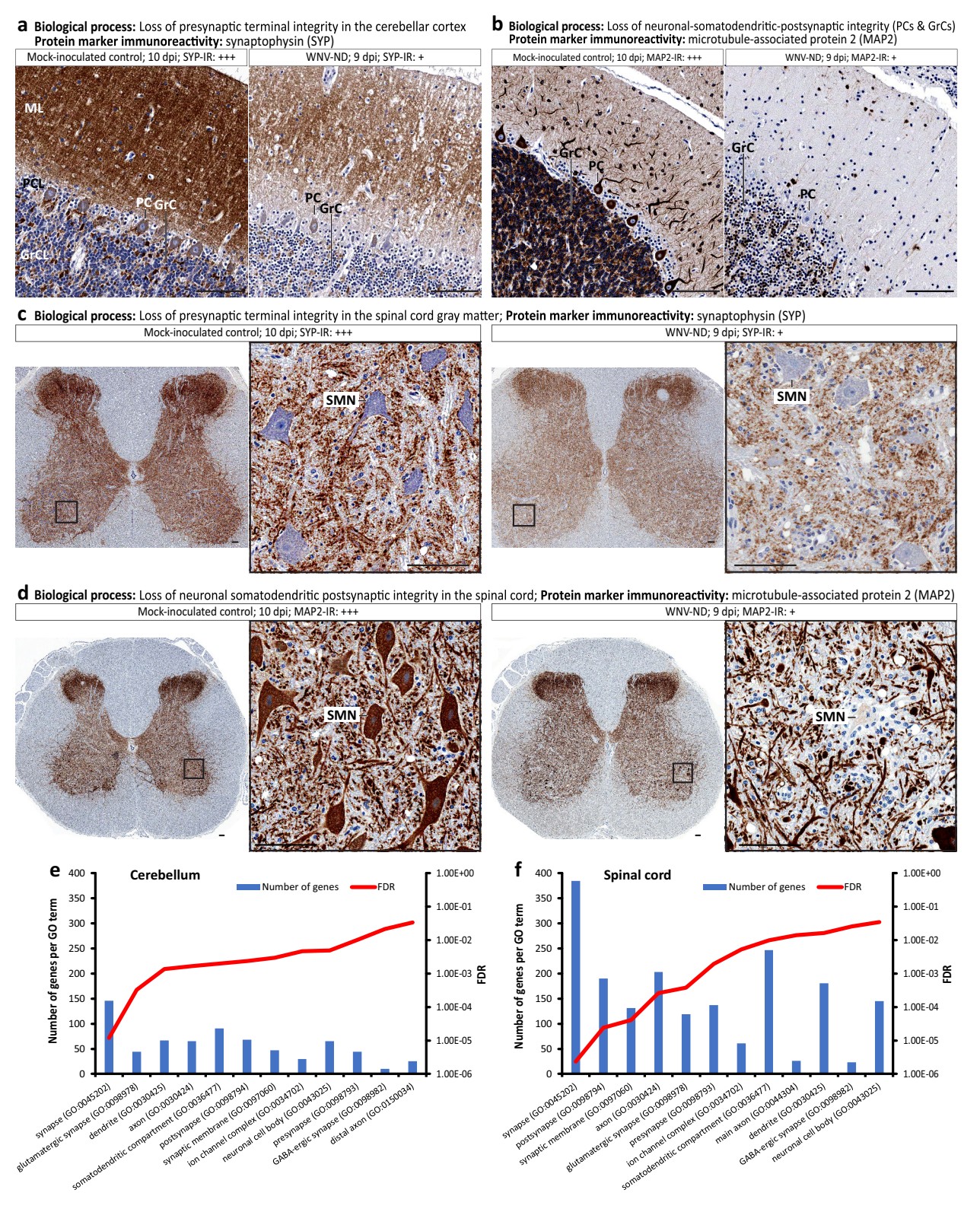

**Figure 8.** Loss of neuronal cell structural organization and function during advanced-symptomatic West Nile virus neurological disease (WNV-ND). (a–d) Representative images illustrate major pathological processes in the cerebellar cortex and spinal cord gray matter (indicated above each panel) by displaying the immunoreactivity (IR) for relevant protein markers (brown) in WNV-infected cerebellum and spinal cord versus mock-inoculated control. Semi-quantitative assessment of the IR is as follows: -, negative; +, minimal; ++; moderate; +++, strong. (e and f) Ranking of neuronal CC GO terms

*Figure 8 continued on next page*

*Figure 8 continued*

based on their enrichment values in the cerebellum (**e**) and spinal cord (**f**) at the advanced-symptomatic stage (9 dpi) of WNV-ND. Plotted for each CC GO term (x-axes) are the number of differentially expressed genes (left y-axes) and FDR-adjusted p-values (right y-axes). ML, molecular layer. PCL, Purkinje cell layer. GrCL, granule cell layer. PC, Purkinje cell. GrC, granule cell. SMN, spinal motor neuron. Scale bars: 100 μm.

displayed neuron-centripetal migration and perineuronal topology (***Figure 6b***). These changes in microglial cell morphology and topology corresponded to the significantly enriched GO terms related to the innate immune response, regulation of microglial cell/macrophage activation, regulation of macrophage chemotaxis, and phagocytosis (***Table 1***).

Immunoreactivity for the lymphocytic protein markers CD20, CD4, and CD8 was detected in the cerebellum and spinal cord in advanced WNV-ND, consistent with infiltration of the CNS by peripheral CNS-extrinsic immune cells (***Figure 6c***). CD20 B cells were detected in the leptomeninges but not in the parenchyma of the cerebellum and only occasionally in perivascular sites in the spinal cord, suggesting limited migration of these cells beyond the initial sites of infiltration. CD4 T cells were detected mostly in the leptomeninges and at perivascular sites in both the cerebellum and spinal cord, and minimally in the parenchyma, also suggesting limited parenchymal migration of these cells. In contrast, numerous CD8 T cells were detected in leptomeningeal, perivascular, and parenchymal locations. Moreover, CD8 T cells displayed a neuron-centripetal migration and perineuronal topology. The infiltration of the CNS by these lymphocytic subtypes and differential regulation of their migration patterns corresponded to the significantly enriched GO terms related to the adaptive immune response and lymphocyte activation/migration (***Table 1***). In addition, the presence of the perivascular infiltration by B cells in the spinal cord gray matter, but not in the cerebellar cortex may be associated with the increased transcriptional regulation of B-cell apoptosis in the spinal cord but not in the cerebellum (***Figure 5c***), thus providing further support to the notion of differential lymphocytic responses to WNV infection in these two CNS regions.

## Subtle coordinated transcriptional shifts correspond to alteration of the structural integrity and function of infected neurons

Based on the identified subtle coordinated shifts in transcriptional regulation of the CNS homeostasis in WMV-ND (***Figure 4e–h***; ***Supplementary file 1*** ['Coordinated transcriptional shifts']), we selected two major BPs and examined WNV-infected tissues by immunohistochemistry, using appropriate protein markers. These processes were: (i) virus infection of specific neuronal cell types; and (ii) disruption of structural integrity and function of infected neurons (***Table 1***).

To identify the specific neuronal cell types supporting WNV replication in the cerebellum and spinal cord, we used brightfield immunohistochemistry for WNV-antigens followed by double immunofluorescent staining for WNV-antigens and appropriate cell type markers. Colorimetric immunoreactivity for WNV-antigens (brown) was detected in Purkinje cells in the cerebellum (***Figure 7a***) and motor neurons in the ventral horns of the spinal cord (***Figure 7c***) in advanced WNV-ND, consistent with typical infection of these neuronal types by flaviviruses. Corresponding to the host response to viral infection, the significantly enriched GO terms were related to the regulation of viral process and viral entry into host cell (***Table 1***).

Strikingly, double immunofluorescent staining for WNV-antigens and a calcium-binding protein calbindin D28k (CALB, protein marker for Purkinje cells) revealed that immunoreactivity for CALB was diminished in Purkinje cells that were WNV-infected (i.e., WNV+++/CALB+ PCs), compared to highly intense CALB-immunoreactivity in non-infected Purkinje cells (i.e., WNV-/CALB+++ PCs) (***Figure 7b***). CALB-immunoreactivity was markedly reduced in the somata of WNV-infected Purkinje cells and almost entirely lost from their dendritic trees extending into the molecular layer of the cerebellar cortex (***Figure 7b***). This finding is consistent with a disturbance of cellular calcium homeostasis and loss of the integrity of somatodendritic compartments of Purkinje cells induced by WNV infection at both protein and gene expression levels (***Table 1***).

Similar to the staining pattern with WNV-antigens and a neuron-specific marker observed in the Purkinje cells, double immunofluorescent staining for WNV-antigens and spinal motor neuron-specific marker choline acetyltransferase (ChAT) revealed that ChAT-immunoreactivity was markedly reduced in the somata and processes of spinal motor neurons that were WNV-infected (i.e., WNV++

+/ChAT+ SMNs), compared to highly intense ChAT-immunoreactivity in non-infected spinal motor neurons (i.e., WNV-/ChAT+++ SMNs) (*Figure 7d*). Strikingly, normally abundant ChAT+++ cholinergic presynaptic C-boutons innervating the somata and proximal dendrites of spinal motor neurons (see normal SMN example in mock-inoculated control) almost completely disappeared from the SNN somata with a few boutons remaining adjacent to their proximal dendrites (*Figure 7d*). Interestingly, focal accumulations of WNV+++ granules in the proximal dendrites of SMNs (postsynaptic sites) could be seen immediately adjacent to the remaining ChAT+ cholinergic presynaptic C-boutons, suggesting trans-synaptic spread and/or local postsynaptic replication of WNV (*Figure 7d*, WNV+++/ChAT+ SMN). These findings are consistent with disruption of the somatodendritic and synaptic structural integrity in WNV-infected SMNs, their neurotransmitter secretion, and afferent innervation (*Table 1*), which collectively would impair their function.

Immunoreactivity for the synaptic marker protein synaptophysin (SYP) was greatly depleted in advanced WNV-ND compared to mock, encompassing all layers of the cerebellar cortex (i.e., molecular layer, Purkinje cell layer, and granule cell layer) (*Figure 8a*) and gray matter of the spinal cord (*Figure 8c*).

Immunoreactivity for microtubule-associated protein 2 (MAP2, a marker for somatodendritic [and thus, postsynaptic] compartments of neurons) was also markedly reduced, confirming the loss of neuronal somatodendritic integrity in Purkinje cells and granule cells in the cerebellum (*Figure 8b*) and spinal motor neurons in the ventral horns of the spinal cord (*Figure 8d*). These findings are consistent with the loss of postsynaptic cellular targets for asymmetric (axodendritic) and symmetric (axosomatic) synapses and disruption of integrity of microtubule cytoskeleton in WNV-infected neurons in the cerebellum and spinal cord (*Table 1*).

Together, these alterations in somatodendritic and synaptic integrity of infected neurons corresponded to the significantly enriched GO terms related to the neuronal cell body, dendrites, microtubule cytoskeleton organization, and synapse (*Table 1*).

Based on these findings, we were interested in identifying a cellular compartment of neurons that was a major target of differential transcriptional regulation in advanced WNV-ND. Ranking the neuronal CC GO terms based on their enrichment values, we identified the synapse as the most significantly dysregulated neuronal cell compartment (*Figure 8e and f*; note the smallest FDR value for the GO CC term synapse (GO:0045202)). This phenomenon was observed in both the cerebellum and spinal cord, albeit the spinal cord had about 2.5 times more dysregulated genes annotated to the GO CC term synapse compared to the cerebellum. Within the synapse, significantly enriched child GO terms included the presynapse (GO:0098793), synaptic membrane (GO:0097060), postsynapse (GO:0098794), and ion channel complex (GO:0034702). Further defining the types of synapses affected in WNV-ND, glutamatergic (i.e., asymmetric, axo-dendritic, and excitatory) synapses (GO:0098978) and GABA-ergic (i.e., symmetric, axo-somatic, and inhibitory) synapses (GO:0098982) emerged as significantly transcriptionally dysregulated in both the cerebellum and spinal cord (*Figure 8e and f*).

Topography of the observed loss of the presynaptic compartments (*Figure 7d*; *Figure 8a and c*) and postsynaptic cellular targets (*Figure 8b and d*) in WNV-ND suggests that (i) putatively affected synapses in the cerebellar cortex may include synapses innervating Purkinje cells (e.g., climbing fiber-PC and parallel fiber-PC [glutamatergic, asymmetric, axo-dendritic, and excitatory] in the molecular layer, Basket cell-PC [GABA-ergic and inhibitory] in the molecular and PC layer, and stellate cell-PC [GABA-ergic and inhibitory] in the molecular layer); and (ii) putatively affected synapses in the spinal cord may include synapses innervating the spinal motor neurons (e.g., cholinergic C-boutons, glutamatergic synapses from descending tracts, and inhibitory synapses from local inhibitory neuron networks) (*Table 1*).

Taken together, these results reveal numerous alterations of the structural integrity and function of infected neurons and suggest that the synapse is a major target of transcriptional dysregulation in WNV-infected cerebellar and spinal neurons, which is supported by evidence of changes in synaptic integrity at the level of protein expression.

## WNV infection disrupts transcriptional regulation of synaptic organization and function

Having identified the synapse as the top transcriptionally dysregulated neuronal cell compartment in advanced WNV-ND, we next asked what specific synaptic subcompartments (hereafter referred as

synaptic location) and BP (hereafter referred as function) were affected. We also sought to infer whether there was a loss or gain of synaptic functions based on the synaptic genes that were upregulated or downregulated during the symptomatic stages of WNV-ND. For this, we used the genomic analysis tool SynGO, a knowledge base that accumulates research about the synapse biology and includes about 3000 expert-curated GO annotations for 1112 synaptic genes (*Koopmans et al., 2019*). SynGO synaptic ontology can be visualized as sunburst diagrams for synaptic location and function (*Figure 9a and b*, respectively). To determine a precise impact of WNV infection on transcriptional regulation of the synapses, we used high stringency SynGO settings, where synaptic gene annotations are based exclusively on published experimental evidence from neuronal biological systems.

SynGO enrichment analysis of the DEGs annotated to the CC GO term synapse (GO:0045202) (*Supplementary file 2*, 'Synaptic genes differentially expressed during WNV-ND') at the advanced-symptomatic stage of WNV-ND revealed extensive transcriptional dysregulation of synapses in both the cerebellum and spinal cord (*Figure 9c–f*). In the cerebellum, the postsynapse was the most significantly affected synaptic subcompartment based on the lowest Q-value (*Figure 9c*; *Table 2*), followed by the presynapse and synaptic cleft, while the most affected synaptic functions were the synapse organization (regulation of synapse assembly/synapse adhesion between pre- and postsynapse), trans-synaptic signaling (modulation of chemical synaptic transmission), processes in presynapse (synaptic vesicle cycle), and processes in postsynapse (regulation of postsynaptic membrane potential/neurotransmitter receptor levels) (*Figure 9d*; *Table 2*). Compared to the cerebellum, the spinal cord had many more synaptic locations (*Figure 9e*; *Table 2*) and functions (*Figure 9f*; *Table 2*) affected and more enriched (including many sublocations and specific synaptic functions). These included synaptic locations such as the presynapse, synaptic membrane, synaptic cleft, and postsynapse; and synaptic functions such as synapse organization, trans-synaptic signaling, processes in pre- and postsynapse, metabolism, and transport. Full results of the SynGO enrichment analysis for all differentially expressed synaptic genes can be found in *Supplementary file 3* ('SynGO enrichment analysis results').

Analysis of upregulated versus downregulated synaptic genes in the cerebellum at the early-symptomatic stage of WNV-ND showed that 80% of synaptic genes were upregulated and 20% were downregulated (*Table 3*). The upregulated synaptic genes were postsynaptic while the downregulated genes were presynaptic. Transcriptional regulation in the spinal cord at this early-symptomatic stage of WNV-ND was characterized by a similar number of upregulated (51.2%) and downregulated (48.8%) synaptic genes, which were annotated to both the presynapse and postsynapse and had functions in synapse organization and chemical synaptic transmission (*Table 3*).

At the advanced-symptomatic stage of WNV-ND, the number of downregulated synaptic genes increased relative to upregulated genes in both the cerebellum and spinal cord, with a higher percentage of downregulated genes in the spinal cord (71.5%). Nonetheless, about one third of the differentially expressed synaptic genes were upregulated at this stage in both the cerebellum and spinal cord (45.9% and 28.5%, respectively). The upregulated and downregulated synaptic genes shared their annotated location and function suggesting a bidirectional dysregulation of transcriptional synaptic homeostasis in WNV-ND, affecting the spinal cord to a higher degree compared to the cerebellum (*Table 3*). Full results of the SynGO enrichment analysis for the upregulated and downregulated synaptic genes can be found in Supplementary File S4 ('SynGO enrichment: up- and downregulated synaptic genes').

Taken together, these findings suggest that WNV infection disrupts transcriptional homeostasis of the CNS synapses and impairs neurotransmission.

## WNV infection induces differential expression of pleiotropic genes with a triple immune, neural, and synaptic topology and functionality

Since functional analysis of genes upregulated at the presymptomatic (3 dpi) stage of WNV-ND showed that some of genes (*MX1* and *STAT1*) have a dual immune-neural functionality (i.e., being pleiotropic), we next sought to determine whether the number of such genes increased as infection progressed. With a focus on the advanced-symptomatic stage of WNV-ND, that had the highest number of dysregulated genes, we identified overlaps in DEGs annotated to the immune or nervous system (hereafter referred as immune or neural DEGs, respectively) (*Figure 10a*).

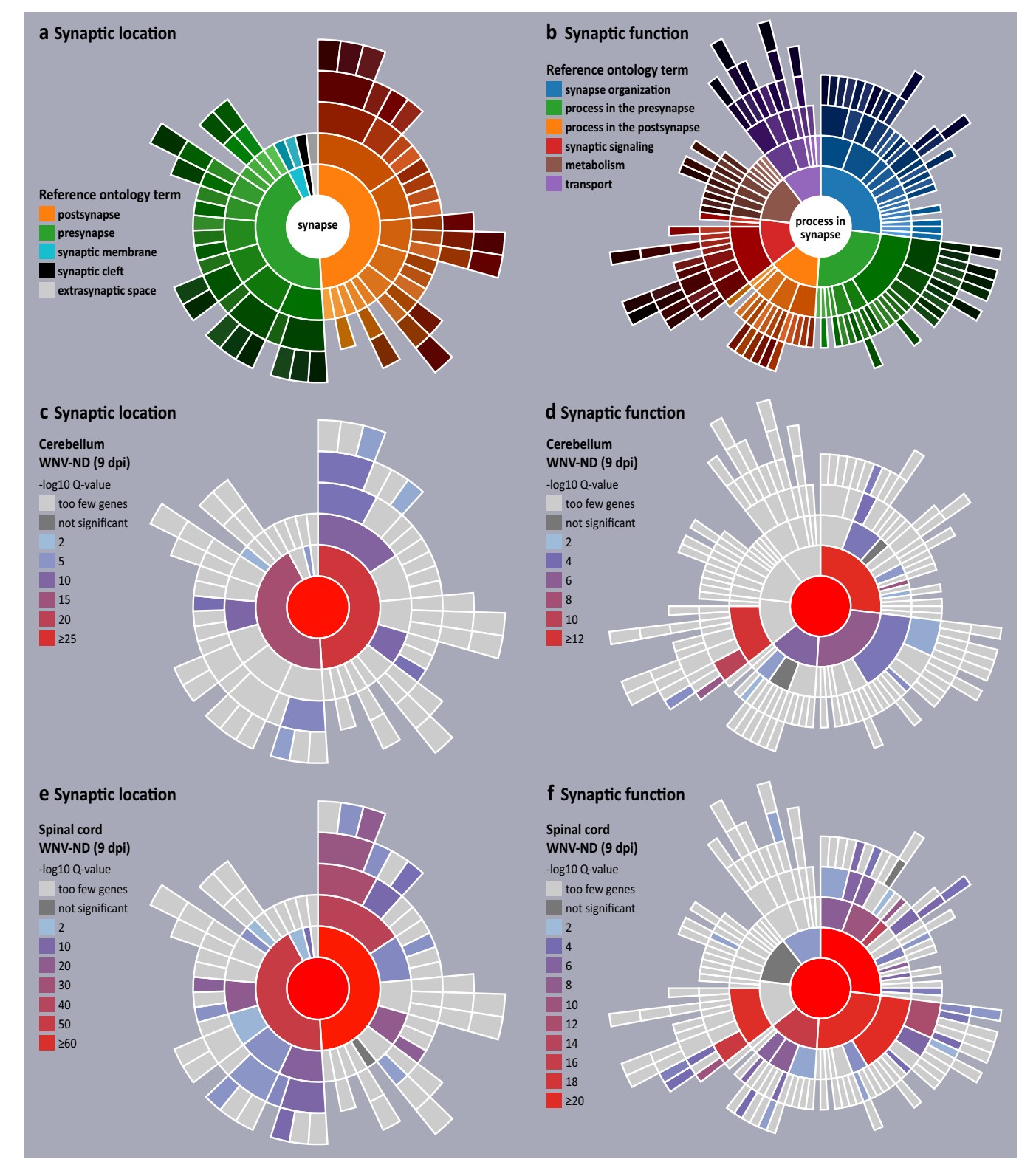

**Figure 9.** Visualization of transcriptional dysregulation of the synaptic organization and function in West Nile virus neurological disease (WNV-ND). (a and b) Reference sunburst diagrams for synaptic location (a) and synaptic function (b) gene ontologies (GOs). The top-level GO terms 'synapse' and 'process in synapse' are at the center of corresponding sunbursts. GO terms representing major synaptic subcompartments (a) or synaptic functions (b) are positioned in the next level from the center, color-coded and shown in the legends. Child GO terms are positioned in the successive rings and

*Figure 9 continued on next page*

*Figure 9 continued*

colored by progressively darkening hues. (**c–f**) Transcriptional synaptic dysregulation in the cerebellum (**c and d**) and spinal cord (**e and f**) is visualized by the overlaying reference sunbursts with a color based on the -log10 Q values (FDR corrected raw p-values) for enriched synaptic GO terms. Specific synaptic GO terms are listed in *Table 2*. Stringent high-level evidence SynGO settings were applied.

Relative to the total number of WNV-induced DEGs, the cerebellum had a higher percentage of immune DEGs (33.1%), compared to the spinal cord (18.6%), even though the absolute number of immune DEGs in the spinal cord was higher. Strikingly, despite more than twice the number of the neural DEGs induced in the spinal cord compared to the cerebellum (642 vs. 252), the percentage of neural DEGs relative to the total number of DEGs was almost identical in these two CNS structures (14.6% vs. 14.5%) (*Figure 10a*). These data suggest that specific CNS regions have differences in immune responses to WNV infection, but more uniform responses related to regulation of neural function.

We next identified 94 and 179 common immune-neural DEGs that were induced in the cerebellum and spinal cord, respectively (*Figure 10a*). This indicates that these DEGs have a dual immune-neural functionality, suggesting that they are pleiotropic (*Radisky et al., 2009*; *Boulanger, 2009*). Common immune-neural as well as exclusively immune or neural DEGs are listed in *Supplementary file 5* ('Venn-diagram results for immune and neural DEGs').

We next asked what neuronal locations and functions were under transcriptional control by the WNV-induced pleiotropic genes. Somatodendritic compartment, neuron projection, and synapse were identified as the main neuronal targets of regulation by pleiotropic genes (ORT; CC GO domain; FDR < 0.05) (*Table 4*). Neuron projection development, regulation of the neuron death, microglial/astroglial cell activation/migration, axon guidance, synapse structure/activity, postsynapse to nucleus signaling, and synapse pruning were identified as the main functions regulated by pleiotropic genes (ORT; BP GO domain; FDR < 0.05) (*Table 4*). The complete functional annotations for the pleiotropic immune-neural genes in WNV-infected cerebellum and spinal cord are provided in *Supplementary file 6* ('ORT: Immune-neural pleiotropic DEGs').

Since functional characterization of WNV-induced pleiotropic immune-neural DEGs suggested that the synapse is one of their major targets in the neuron, we next determined the overlap between these pleiotropic DEGs and WNV-induced synaptic DEGs identified (*Supplementary file 2*, 'Synaptic genes differentially expressed during WNV-ND'). To compare these genes, we focused on the advanced-symptomatic stage of WNV-ND in the spinal cord since it had the highest number of genes in both groups. A Venn diagram comparison identified an overlap containing 36 immune-neural-synaptic pleiotropic DEGs, of which 19 were upregulated and 17 were downregulated (*Figure 9b*; *Supplementary file 7*, 'Immune-neural-synaptic pleiotropic genes'). Functional GO enrichment analysis of the upregulated genes in this overlap indicated that they had multiple roles in the positive regulation of the following functions: (i) immune/defense response to virus; (ii) nervous system development (neurogenesis/gliogenesis); and (iii) neuronal cell body/projections/synapses-specific processes (*Figure 10c and d*). These processes were also significantly enriched for the downregulated pleiotropic genes, with a trend of less significant (compared to upregulated genes) enrichment of immune processes and more significant enrichment of processes related to the organization, structure, and/or activity of the neuronal compartments (*Figure 10e and f*). Full results of functional GO enrichment analysis for the upregulated and downregulated pleiotropic immune-neural-synaptic genes are provided in *Supplementary file 8* ('gProfiler: Immune-neural-synaptic pleiotropic DEGs').

Interestingly, interrogation of specific pleiotropic genes showed that the gene *C1QA* that encodes a major constituent of the complement system subcomponent C1, was upregulated and mapped to significantly enriched GO terms associated with (i) immune/defense response to virus infection, (ii) glial cell response, (iii) neuron differentiation, and (iv) synapse (*Supplementary file 8*, gProfiler: Immune-neural-synaptic pleiotropic DEGs'), as well as synaptic pruning (*Table 4*).

Since the synapse emerged as the neuronal cell compartment most significantly enriched for pleiotropic genes (*Figure 10c–f*), we applied synaptic GO enrichment analysis (SynGO) and found that both up- and downregulated genes had functionality in synapse organization and topology (i.e., localization of the gene product functions) at the presynapse and postsynapse (*Figure 10g*). Specifically, the GO term of chemical synaptic transmission was significantly enriched for upregulated

**Table 2.** Enriched GO terms for the synaptic genes dysregulated in the cerebellum and spinal cord in WNV-ND.

| GO term ID | GO domain | GO term name - hierarchical structure | Q-value Spinal cord | Cerebellum |
|---|---|---|---|---|
| **GO:0045202** | CC | **synapse** | **7.13E-113** | **3.56E-35** |
| GO:0097060 | CC | ├── synaptic membrane | 6.07E-03 | |
| GO:0098793 | CC | ├── presynapse | 4.20E-51 | 4.19E-17 |
| GO:0099523 | CC | │ ├── presynaptic cytosol | 7.07E-05 | 1.79E-03 |
| GO:0048786 | CC | │ ├── presynaptic active zone | 1.52E-15 | |
| GO:0048787 | CC | │ │ └── presynaptic active zone membrane | 3.25E-11 | 1.08E-06 |
| GO:0098833 | CC | │ ├── presynaptic endocytic zone | 6.87E-03 | |
| GO:0008021 | CC | │ ├── synaptic vesicle | 6.06E-05 | |
| GO:0098992 | CC | │ ├── neuronal dense core vesicle | 4.44E-03 | |
| GO:0042734 | CC | │ ├── presynaptic membrane | 4.46E-21 | 3.52E-10 |
| GO:0043083 | CC | ├──synaptic cleft | 5.66E-07 | 2.40E-05 |
| GO:0098794 | CC | ├── postsynapse | 5.38E-80 | 4.68E-24 |
| GO:0099524 | CC | │ ├── postsynaptic cytosol | 1.61E-02 | |
| GO:0099571 | CC | │ ├── postsynaptic cytoskeleton | 2.60E-05 | |
| GO:0099572 | CC | │ ├── postsynaptic specialization | 4.46E-44 | 4.94E-11 |
| GO:0014069 | CC | │ │ ├── postsynaptic density | 1.73E-34 | 3.60E-08 |
| GO:0045211 | CC | │ ├── postsynaptic membrane | 1.12E-21 | 6.60E-10 |
| SYNGO:postsyn_ser | CC | │ └── postsynaptic SER | 3.30E-04 | |
| **SYNGO:synprocess** | BP | **process in the synapse** | **1.30E-87** | **2.74E-35** |
| **SYNGO:presynprocess** | BP | ├── **process in the presynapse** | **4.96E-21** | **4.11E-07** |
| GO:0099509 | BP | │ ├── regulation of presynaptic cytosolic calcium levels | 7.25E-04 | |
| GO:0099504 | BP | │ ├── synaptic vesicle cycle | 5.53E-21 | 2.01E-04 |
| GO:0016079 | BP | │ │ ├── synaptic vesicle exocytosis | 9.78E-13 | 3.58E-03 |
| GO:0099502 | BP | │ │ │ ├── calcium-dependent activation of synaptic vesicle fusion | 4.74E-04 | |
| GO:2000300 | BP | │ │ │ ├── regulation of synaptic vesicle exocytosis | 1.82E-04 | |
| GO:0016082 | BP | │ │ │ └── synaptic vesicle priming | 4.88E-05 | |
| GO:0048488 | BP | │ │ └── synaptic vesicle endocytosis | 6.15E-06 | |
| **SYNGO:postsynprocess** | BP | ├── **process in the postsynapse** | **5.44E-17** | **8.45E-06** |
| GO:0099566 | BP | │ ├── regulation of postsynaptic cytosolic calcium levels | 2.92E-03 | |
| GO:0060078 | BP | │ ├── regulation of postsynaptic membrane potential | 2.75E-07 | 1.63E-03 |
| GO:0099072 | BP | │ ├── regulation of postsynaptic membrane neurotransmitter receptor levels | 6.34E-08 | 1.10E-02 |
| GO:0099645 | BP | │ │ ├── neurotransmitter receptor localization to postsynaptic specialization membrane | 1.70E-05 | |
| GO:0099149 | BP | │ │ └── regulation of postsynaptic neurotransmitter receptor endocytosis | 9.33E-05 | |
| **GO:0099537** | BP | │ ├── **trans-synaptic signaling** | **1.47E-20** | **8.11E-13** |
| GO:0007268 | BP | │ │ └── chemical synaptic transmission | 1.11E-17 | 1.43E-10 |
| GO:0050804 | BP | │ │ ├── modulation of chemical synaptic transmission | 8.44E-11 | 1.63E-07 |
| GO:0099171 | BP | │ │ │ ├── presynaptic modulation of chemical synaptic transmission | 9.43E-05 | 7.08E-04 |
| GO:0099170 | BP | │ │ └── postsynaptic modulation of chemical synaptic transmission | 2.89E-04 | |
| **GO:0050808** | BP | ├── **synapse organization** | **4.53E-45** | **4.99E-14** |
| GO:0099173 | BP | │ ├── postsynapse organization | 5.47E-15 | 1.91E-02 |
| GO:0099010 | BP | │ │ └── modification of postsynaptic structure | 1.92E-05 | |
| GO:0099181 | BP | │ │ ├── structural constituent of presynapse | 1.46E-06 | |

*Table 2 continued on next page*

*Table 2 continued*

| GO term ID | GO domain | GO term name - hierarchical structure | Q-value Spinal cord | Cerebellum |
|---|---|---|---|---|
| GO:0098882 | BP | \| \| \| ├─ structural constituent of active zone | 9.80E-06 | |
| GO:0099186 | BP | \| \| └─ structural constituent of postsynapse | 2.40E-03 | |
| GO:0099560 | BP | \| ├─ synapse adhesion between pre- and post-synapse | 8.52E-06 | 4.79E-03 |
| GO:0007416 | BP | \| ├─ synapse assembly | 1.75E-10 | 2.01E-04 |
| GO:0099054 | BP | \| \| ├─ presynapse assembly | 8.02E-03 | |
| GO:0051963 | BP | \| \| ├─ regulation of synapse assembly | 4.99E-07 | 1.19E-04 |
| GO:1905606 | BP | \| \| \| ├─ regulation of presynapse assembly | 1.37E-04 | 2.76E-04 |
| GO:0097107 | BP | \| \| \| └─ postsynaptic density assembly | 1.03E-02 | |
| GO:0060074 | BP | \| ├─ synapse maturation | 6.28E-05 | |
| GO:0099188 | BP | \| └─ postsynaptic cytoskeleton organization | 2.89E-04 | 1.46E-03 |
| **SYNGO:metabolism** | BP | ├─ metabolism | **1.69E-02** | |
| SYNGO: catabolic_postsynapse | BP | \| \| └─ protein catabolic process at postsynapse | 2.40E-03 | |
| **SYNGO:transport** | BP | └─ transport | **1.48E-03** | |
| GO:0098887 | BP | \| └─ neurotransmitter receptor transport, endosome to postsynaptic membrane | 2.40E-03 | |

Note: The top-level GO terms 'synapse' and 'process in the synapse' and major successive terms for synaptic subcompartments and processes are highlighted for the cerebellum and spinal cord at the advanced-symptomatic stage of WNV-ND. CC, cellular component; BP, biological process.

pleiotropic genes, while the GO term of protein catabolic process at the postsynapse was significantly enriched for downregulated pleiotropic genes (*Figure 10g*). A complete list of the enriched synaptic GO terms is provided in *Supplementary file 9* ('Pleiotropic DEGs: Synaptic topology and functionality').

Taken together, these results indicate that WNV infection of neurons induces differential expression of pleiotropic genes that have multiple functionalities. Strikingly, we found that in addition to their expected role in immune/defense responses during WNV-ND, these genes also regulated distinct functions in neurons and their synapses. This suggests that WNV infection of neurons disrupts transcriptional homeostasis of the immune-neural-synaptic axis with possible off-target effects of virus-induced immune responses on neural function.

## Validation of WNV-ND transcriptome

We performed the validation of the WNV-ND transcriptome in NHPs at the level of select transcript expression by the qPCR. The genes for qPCR validation were selected based on their involvement in the regulation of several BPs that were found to be affected in this study: cellular calcium ion homeostasis (CALB1); excitatory synapses (GRIA1); inhibitory synapses (GABRA2 and GABBR1); perineuronal/perisynaptic extracellular matrix (TNC); astrocyte activation (GFAP); complement activation (C3); antigen processing and presentation via MHC class I (B2M); and immune cell chemotaxis/migration (CXCL10). qPCR analysis of these select transcripts had confirmed their significant (one-way ANOVA; $p < 0.05$) dysregulation during WNV infection in the cerebella of NHPs (*Supplementary file 10*, 'WNV-ND transcriptome validation'). In addition, a linear regression analysis returned the strong and significant correlations between the expression values for these select genes as determined by microarray or qPCR at the symptomatic stages of WNV-ND (early-symptomatic stage: $R^2 = 0.998$, $p < 0.05$; advanced-symptomatic stage: $R^2 = 0.987$, $p < 0.05$). Taken together, these results validate the transcriptome data by the qPCR and provide a strong additional support to the functional genomics of the WNV-ND in NHPs reported here.

**Table 3.** Enriched GO terms for the upregulated or downregulated synaptic genes in WNV-infected cerebellum and spinal cord.

| | Number of upregulated genes (percent) | Enriched GO terms | Number of downregulated genes (percent) | Enriched GO terms |
|---|---|---|---|---|
| *Early-symptomatic* | | | | |
| Cerebellum | 24 (80%) | postsynaptic specialization | 6 (20%) | presynapse |
| Spinal cord | 44 (51.2%) | presynapse; postsynapse; postsynaptic specialization; synapse organization; synapse assembly; chemical synaptic transmission | 42 (48.8%) | postsynapse; postsynaptic specialization; synapse organization; regulation of postsynaptic membrane neurotransmitter receptor levels; modulation of chemical synaptic transmission |
| *Advanced-symptomatic* | | | | |
| Cerebellum | 67 (45.9%) | synapse organization; regulation of synapse assembly; presynaptic active zone membrane; presynaptic modulation of chemical synaptic transmission; postsynaptic specialization; metabolism | 79 (54.1%) | regulation of synapse organization; regulation of presynapse assembly; presynaptic cytosol; presynaptic active zone membrane; regulation of presynaptic membrane potential; postsynaptic actin cytoskeleton organization; postsynaptic specialization; regulation of postsynaptic membrane neuro-transmitter receptor levels; regulation of postsynaptic membrane potential; synapse adhesion between pre- and postsynapse; regulation of synaptic vesicle cycle; modulation of chemical synaptic transmission |
| Spinal cord | 109 (28.5%) | synapse organization; regulation of synapse assembly; synaptic vesicle membrane; synapse adhesion between pre- and post-synapse; synaptic cleft; presynaptic active zone membrane; presynaptic modulation of chemical synaptic transmission; postsynaptic specialization; synaptic vesicle cycle; modulation of chemical synaptic transmission | 274 (71.5%) | regulation of synapse organization; synaptic vesicle membrane; neuronal dense core vesicle; regulation of synaptic vesicle cycle; synaptic vesicle neurotransmitter loading; synaptic vesicle priming; synapse adhesion between pre- and postsynapse; synaptic cleft; structural constituent of presynapse; presynaptic cytosol; presynaptic active zone cytoplasmic component; presynaptic active zone membrane; presynaptic endocytic zone; regulation of presynapse assembly; regulation of presynaptic cytosolic calcium levels; regulation of presynaptic membrane potential; presynaptic modulation of chemical synaptic transmission; structural constituent of postsynapse; regulation of postsynapse organization; postsynaptic cytosol; regulation of modification of postsynaptic actin cytoskeleton; modification of postsynaptic structure; postsynaptic specialization assembly; regulation of postsynaptic cytosolic calcium levels; regulation of calcium-dependent activation of synaptic vesicle fusion; protein catabolic process at postsynapse; regulation of postsynaptic membrane neurotransmitter receptor levels; regulation of postsynaptic neurotransmitter receptor activity; regulation of postsynaptic membrane potential; postsynaptic modulation of chemical synaptic transmission; transport; metabolism |

## Discussion

Severe disease of the CNS due to infection with flaviviruses is rare, but the consequences for neural function can be devastating (reviewed in *Maximova and Pletnev, 2018*). Pathogenesis and outcome of CNS diseases caused by flaviviruses are complex and involve an interplay of at least three components: (i) virus properties (i.e., virulence, ability to establish productive replication in neurons, and spread by axonal transport); (ii) host defense responses and/or viral evasion of these responses; and (iii) changes in neural function with ensuing specific neurological impairments. Although we now possess a substantial body of knowledge about flavivirus biology and host immune responses to infection (reviewed in *Pierson and Diamond, 2020*), including immune responses that develop specifically within the CNS (reviewed in *Maximova and Pletnev, 2018*), our understanding of how

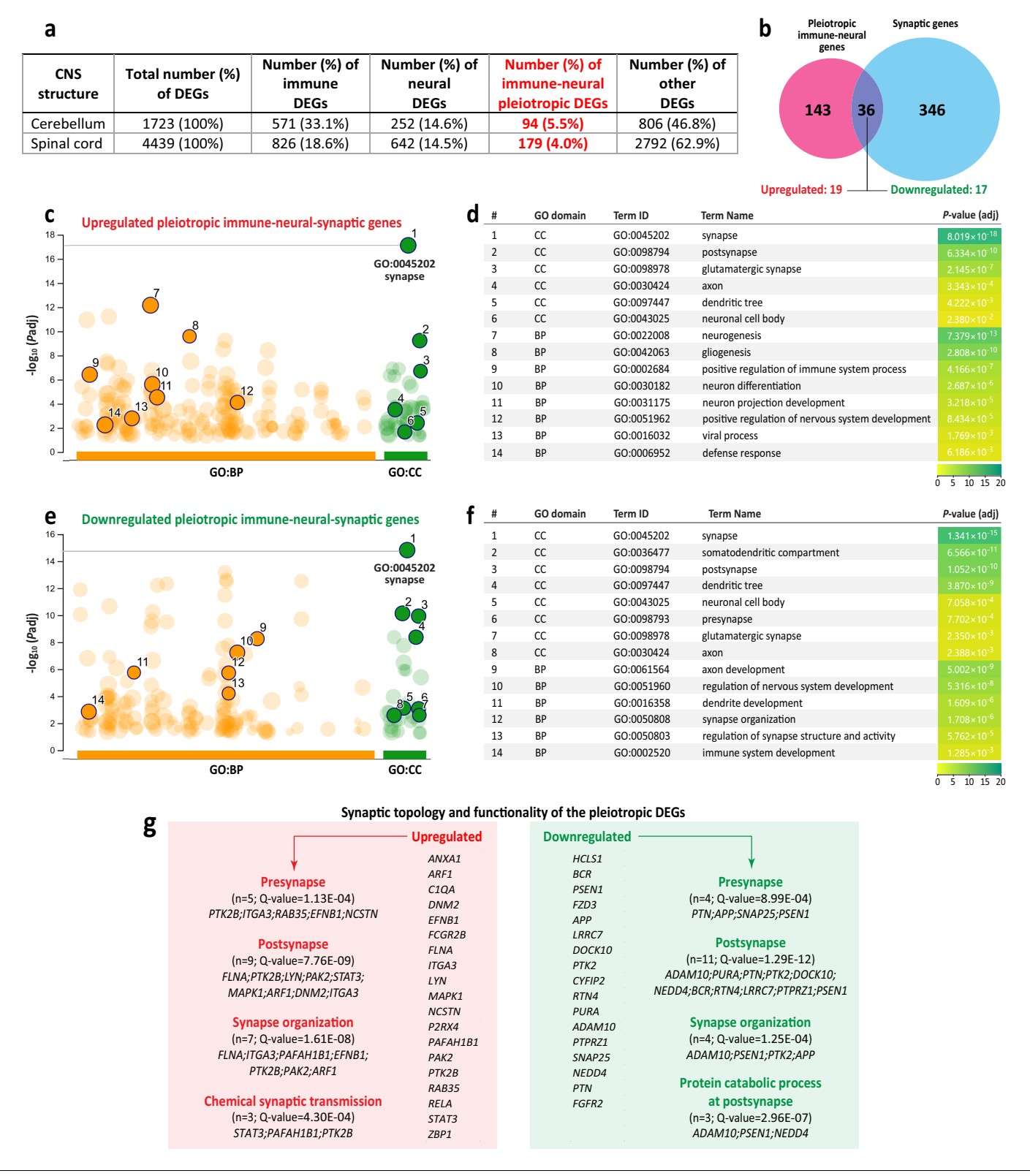

**Figure 10.** Identification and functional analysis of pleiotropic immune-neural-synaptic genes induced by West Nile virus (WNV) infection. (**a**) Distribution of differentially expressed genes (DEGs) annotated to the immune or nervous system, or both (immune-neural pleiotropic DEGs; highlighted in red), relative to the total number of DEGs in indicated central nervous system (CNS) structures at the advanced-symptomatic stage of WNV neurological disease (WNV-ND). (**b**) Venn diagram shows the overlap (n = 36) between the immune-neural pleiotropic and synaptic DEGs in the

*Figure 10 continued on next page*

Figure 10 continued

spinal cord at the advanced-symptomatic stage of WNV-ND (numbers of the up- or downregulated genes in the overlap are indicated). (c–f) Select significantly enriched BP and CC GO terms (orange and green solid circles, respectively) and their respective statistical data (identified using gProfiler) for the upregulated (c and d) or downregulated (e and f) immune-neural-synaptic pleiotropic DEGs. (g) Gene symbols, significantly enriched SynGO terms, and respective statistical data for immune-neural-synaptic pleiotropic DEGs.

they directly or indirectly influence neural function is incomplete. Here, we aimed to deconvolute the complex changes in CNS physiology that occur during flavivirus infection by examining differential regulation of BPs related to the immune and nervous systems in the NHP model of WNV-ND. We chose to focus on the cerebellum and spinal cord since these CNS structures appear to be among the most affected in both humans with WNV neuroinvasive disease (*Kleinschmidt-DeMasters and Beckham, 2015*; *Sejvar, 2016*; *Lenka et al., 2019*; *Omalu et al., 2003*; *Cushing et al., 2004*; *Guarner et al., 2004*; *Armah et al., 2007*; *Hart et al., 2014*) and NHP model of WNV-ND (*Maximova et al., 2016*; *Maximova et al., 2014*), causing ataxia and flaccid paralysis, respectively.

CNS neurons are terminally differentiated cells and cannot be replenished, thus virus infection of neurons present a challenge for both the nervous and immune system to coordinate virus clearance while protecting neurons and preserving neural function. While innate immune responses in the CNS are common during virus infection (*Maximova and Pletnev, 2018*; *Griffin, 2011*; *Griffin, 2003*), despite the concept of immune privilege status of the CNS (*Ransohoff and Brown, 2012*), these responses need to be tightly controlled, since excessive or chronic inflammation can be harmful and detrimental to neural function (*Griffin, 2011*; *Ransohoff and Brown, 2012*; *Ransohoff and Cardona, 2010*). Here we introduce the concept that virus infection of CNS neurons disrupts the immune-neural axis homeostasis due to activation of pleiotropic genes that can regulate both innate immunity and neural function. Pleiotropy describes a concept where one gene and its encoded protein can control disparate, apparently unrelated BPs (*Radisky et al., 2009*; *Boulanger, 2009*). This concept is especially intriguing in the context of regulation of critically important neural functions. Many proteins with established functions in the immune system are also expressed in the developing and adult nervous system. Remarkably, some pleiotropic proteins such as proinflammatory cytokines and proteins in the complement system and major histocompatibility complex are essential for the establishment, organization, function, and removal of synapses between neurons (*Boulanger, 2009*). We found that WNV infection of neurons in the cerebellar and spinal neural networks induces differential regulation (i.e., both up- and downregulation) of close to 200 pleiotropic genes that possess dual immunological and neurological topology and functionality. Strikingly, among the affected neuronal cell compartments, the synapses, both excitatory (glutamatergic) and inhibitory (GABA-ergic), emerged in this study as the most significantly transcriptionally dysregulated compartments. This was supported by the evidence of reduction in expression of the pan-presynaptic protein marker synaptophysin in the cerebellar and spinal motor circuitries. In addition, we identified a set of the pleiotropic genes with a triple topology and functionality in the immune system, nervous system in general (including functions of neuronal cell components and glial cells), and specifically, in neuron-to-neuron synapses (i.e., immune-neural-synaptic pleiotropic genes; n = 36). Focused spatial and functional interrogation of these pleiotropic genes that became possible with a recent release of a comprehensive synaptic biology knowledgebase SynGO (*Koopmans et al., 2019*), suggested their involvement in the regulation of synapse (including pre- and post-synapses) organization, chemical synaptic transmission, and protein catabolic processes. This builds upon our conclusion that transcriptional responses to WNV infection result in altered regulation of the organization and function of synapses. Many immune molecules in the neuron-microglia-astrocyte-T-cell axes have been implicated in synaptic pruning during development or pathological elimination/loss of synapses during disease, depending on the neuronal network (*Perry and O'Connor, 2008*; *Eroglu and Barres, 2010*; *Verkhratsky and Nedergaard, 2014*; *Chung et al., 2013*; *Tröscher et al., 2019*; *Kreutzfeldt et al., 2013*; *Di Liberto et al., 2018*; *Stephan et al., 2012*). Synaptic elimination in the hippocampus by a complement-microglial axis (*Vasek et al., 2016*) with assistance of T cells (*Garber et al., 2019*) has also been proposed as an underlying mechanism of flavivirus-associated cognitive dysfunction in a murine model. Our data in NHPs also support a significant role of the neuron-microglia-astrocyte-T-cell axes in the pathogenesis of WNV-ND at the gene and protein expression levels. Furthermore, the identification of the complement system *C1QA* gene (which was

**Table 4.** Neuronal cell compartments and neural functions controlled by West Nile virus (WNV)-induced immune-neural pleiotropic differentially expressed genes (DEGs).

| GO domain | GO term | No. of genes | Gene symbols |
|---|---|---|---|
| Cerebellum | | | |
| CC | somatodendritic compartment (GO:0036477) | 19 | MBP, HCLS1, ITGA4, PTX3, FCGR2B, ZBP1, CCR2, CYFIP1, CTSL2, CIB1, AIF1, SOS1, ANXA3, CD3E, S100B, FLNA, MAPK1, ITGA1, TGFB2 |
| CC | neuron projection (GO:0043005) | 19 | MBP, EPHA2, HCLS1, ITGA4, FCGR2B, ZBP1, VIM, CCR2, BCL11B, CYFIP1, CTSL2, CIB1, ANXA3, CD3E, FLNA, MAPK1, ITGA1, TGFB2, ODZ1 |
| CC | postsynapse (GO:0098794) | 11 | JAK2, HCLS1, C1QA, FCGR2B, ZBP1, CYFIP1, STAT3, SOS1, CD3E, FLNA, MAPK1 |
| BP | neuron projection development (GO:0031175) | 25 | EPHA2, JAK2, IL6, HCLS1, ITGA4, ADM, RHOG, SHC1, BCL11B, CYFIP1, SDC4, LST1, DOK2, CXCR4, SEMA4A, CSF1R, SOS1, CD3E, S100B, HMGB1, GATA3, MAPK1, CXCL12, ITGA1, NCKAP1L |
| BP | positive regulation of neuron death (GO:1901216) | 13 | PTX3, C1QA, FCGR2B, IFNG, BAX, TYROBP, TNF, BCL2L11, GRN, TLR4, ITGA1, TGFB2, CCL3 |
| BP | microglial cell activation (GO:0001774) | 10 | JAK2, C1QA, IFNG, TLR2, TYROBP, TNF, FPR2, AIF1, GRN, C5AR1 |
| BP | astrocyte activation (GO:0048143) | 7 | C1QA, IFNG, TNF, FPR2, IL1B, GRN, C5AR1 |
| BP | glial cell migration (GO:0008347) | 4 | P2RY12, CCL2, TGFB2, CCL3 |
| BP | axon guidance (GO:0007411) | 12 | EPHA2, RHOG, SHC1, BCL11B, CYFIP1, DOK2, CXCR4, CSF1R, SOS1, GATA3, MAPK1, CXCL12 |
| BP | postsynapse to nucleus signaling pathway (GO:0099527) | 3 | JAK2, STAT3, RELA |
| BP | regulation of synapse structure or activity (GO:0050803) | 8 | HCLS1, FCGR2B, TLR2, SEMA4D, CYFIP1, TNF, SLC7A11, SEMA4A |
| BP | positive regulation of glutamate receptor signaling pathway (GO:1900451) | 3 | IFNG, CCR2, CCL2 |
| BP | synapse pruning (GO:0098883) | 2 | C1QA, C3 |
| Spinal cord | | | |
| CC | somatodendritic compartment (GO:0036477) | 39 | REG1A, HSP90AA1, HCLS1, DAB2IP, DHX36, ITGA4, PTX3, FCGR2B, ZBP1, CX3CL1, BCR, HSP90AB1, PTK2B, RARA, FZD3, PSEN1, CCR2, APP, CASP3, PAFAH1B1, DOCK10, ALCAM, ARHGEF7, CTSL2, CIB1, AIF1, PTK2, P2RX4, RTN4, KIF5B, PURA, ADAM10, SNAP25, DNM2, FLNA, MAPK1, NEDD4, BECN1, ITGA1 |
| CC | neuron projection (GO:0043005) | 46 | REG1A, EPHA2, HSP90AA1, HCLS1, DAB2IP, DHX36, ITGA4, FCGR2B, ZBP1, CX3CL1, BCR, HSP90AB1, PTK2B, RARA, FZD3, PSEN1, VIM, CCR2, APP, LRRC7, BSG, PAFAH1B1, DOCK10, ALCAM, ARHGEF7, CTSL2, CIB1, ARF1, PTK2, ITGA3, CYFIP2, P2RX4, KIF5B, PURA, ADAM10, NCAM1, SNAP25, DNM2, FLNA, MAPK1, NEDD4, BECN1, ITGA1, CEP290, ODZ1, BRAF |
| CC | axon (GO:0030424) | 25 | REG1A, HSP90AA1, DAB2IP, DHX36, ITGA4, ZBP1, BCR, HSP90AB1, PTK2B, FZD3, PSEN1, APP, LRRC7, BSG, PAFAH1B1, ALCAM, CIB1, ITGA3, P2RX4, KIF5B, ADAM10, SNAP25, DNM2, FLNA, MAPK1 |
| CC | synapse (GO:0045202) | 37 | HCLS1, C1QA, FCGR2B, ZBP1, RAB35, BCR, PTK2B, FZD3, PSEN1, FZD3, NCSTN, APP, LRRC7, STAT3, PAFAH1B1, DOCK10, LYN, ARF1, PTK2, ITGA3, CYFIP2, EFNB1, RELA, P2RX4, RTN4, PURA, ADAM10, PTPRZ1, SNAP25, DNM2, FLNA, MAPK1, NEDD4, PTN, FGFR2, PAK2, ANXA1 |
| CC | glutamatergic synapse (GO:0098978) | 14 | BCR, PTK2B, FZD3, STAT3, LYN, ARF1, RELA, PURA, ADAM10, SNAP25, DNM2, FLNA, NEDD4, PAK2 |
| BP | neuron projection development (GO:0031175) | 50 | EPHA2, HSP90AA1, HCLS1, IL6, DAB2IP, PIK3R1, ITGA4, SLC11A2, HES1, ADM, JUN, RAB35, HSP90AB1, SEC24B, PTK2B, FZD3, APP, SHC1, PTPN11, CASP3, LST1, BSG, PAFAH1B1, DOK2, DOCK10, ALCAM, LYN, RPS6KA5, PTK2, EIF2AK4, SRF, CYFIP2, SEMA4A, EFNB1, CSF1R, RTN4, KIF5B, PTPRZ1, NCAM1, HMGB1, DNM2, GATA3, MAPK1, CXCL12, NEDD4, PTN, ITGA1, STK4, FGFR2, PAK2 |
| BP | microglial cell activation (GO:0001774) | 14 | TREM2, C1QA, IFNGR1, JUN, IFNG, CX3CL1, TLR2, APP, TYROBP, TNF, FPR2, AIF1, GRN, C5AR1 |
| BP | astrocyte activation (GO:0048143) | 11 | TREM2, C1QA, IFNGR1, IFNG, PSEN1, APP, TNF, FPR2, IL1B, GRN, C5AR1 |
| BP | regulation of glial cell migration (GO:1903975) | 8 | TREM2, CX3CL1, CCR2, GPR183, CSF1, P2RY12, P2RX4, CCL3 |
| BP | regulation of synapse structure or activity (GO:0050803) | 17 | HCLS1, DAB2IP, DHX36, FCGR2B, TLR2, FZD3, APP, SEMA4D, TNF, PAFAH1B1, SLC7A11, PTK2, SEMA4A, ADAM10, DNM2, NEDD4, PTN |
| BP | synapse pruning (GO:0098883) | 3 | TREM2, C1QA, CX3CL1 |

upregulated) as the immune-neural-synaptic pleiotropic gene in this study supports a growing evidence of the multiple roles of its protein product in the immune system and in elimination of synapses during development and disease (*Perry and O'Connor, 2008*). We propose a scenario in which changes in expression of pleiotropic genes that are intended to activate and maintain immune

responses to virus infection would have unintended and potentially devastating off-target effects on neuron-to-neuron synapses and chemical neurotransmission.

Our findings were facilitated by a functional enrichment approach that allows detection of modest but coordinated changes in expression of groups of functionally related genes that may elude other methods (*Mi et al., 2019*). A similar approach has been used to detecting subtle but important coordinated changes in gene expression associated with complex human disorders that otherwise may be overlooked (*Mootha et al., 2003*). By using this approach, we were able to detect subtle coordinated shifts in regulation of the nervous system that were accompanied by disruption of the integrity and function of cellular compartments in infected neurons, especially their synapses. While these shifts are subtle, they may have an enormous impact, especially on the regulation of such crucial function as neural function. Consistent with this, we found that WNV infection disrupts transcriptional homeostasis of the cerebellar and spinal synapses, affecting the regulation of synapse organization, chemical synaptic transmission, metabolism, and transport. WNV-infection-induced upregulated and downregulated synaptic genes shared their topology and functionality, suggesting a bidirectional transcriptional dysregulation of synapse biology. These changes offer a mechanistic explanation for WNV-induced impairment of motor functions such as ataxia, tremor, and limb weakness/paralysis in our NHP model (*Maximova et al., 2014*) and in humans with WNV-ND (*Lenka et al., 2019*).

In support of the important role of the dysregulation of synapse biology with ensuing impairment of neurotransmission as the leading mechanisms underlying pathogenesis and neurological presentations of WNV-ND in primate hosts (NHPs and humans), we also provide evidence of strict transcriptional regulation of the cell death processes in neurons. In fact, it appears that at the advanced-symptomatic stage of WNV-ND, the magnitude of regulation of cell death processes in neurons does not exceed that in lymphocytes, suggesting that neurons are not a major cell type targeted by increased transcriptional regulation of cell death processes. Instead, it appears that the major cell types with increased regulation of cell death processes were lymphocytes (T and B cells that are expected to infiltrate the CNS parenchyma as a part of the adaptive immune response to viral infection [*Maximova and Pletnev, 2018*]). Thus, a closer examination of transcriptional regulation of cell death appears to support the lack of extensive activation of cell death pathways in neurons, especially when compared to other cell types in WNV-infected CNS (i.e., infiltrating T and B cells). In addition, a CNS-region-specific activation of the programmed cell death in infiltrated lymphocytes suggests the initiation of the resolution phase of the acute cellular inflammatory responses in WNV-infected CNS. This is consistent with our previously published results with the NHP model for other flavivirus infections of the CNS (*Maximova et al., 2009*), where we showed that the perivascular infiltrated lymphocytes rather than neurons were undergoing apoptosis.

Since WNV can spread trans-synaptically (*Maximova et al., 2016*), changes in synaptic homeostasis identified in this study may indicate either a direct damaging impact of infection on neurotransmission or an attempt by the host to arrest virus dissemination and compensate for changes in function. More studies will be required to understand to what extent these synaptic changes represent pathological, protective, and/or compensatory mechanisms, as well as whether they are reversible. This may guide a selection of potential therapeutic targets. For instance, our data implicate a disturbance in transcriptional regulation of glutamatergic (excitatory) and GABA-ergic (inhibitory) synapses, suggesting an imbalance between excitatory and inhibitory neurotransmission in the pathogenesis of WNV-ND. If confirmed, this excitatory-inhibitory imbalance could potentially be pharmacologically targeted with existing drugs as a symptomatic treatment (e.g., with medications increasing GABA-ergic neurotransmission, which are effective in symptomatic treatment of essential tremor, a disease affecting the Purkinje cells in the cerebellum [*Louis, 2016*], and in this respect, similar to WNV-ND).

Intriguingly, in our intracerebral NHP model of WNV-ND, transcriptional changes were detected earlier in the lumbar spinal cord, the CNS region most remote from the site of virus inoculation (thalamus), compared to the cerebellum. The changes in gene expression also became much stronger in the spinal cord, compared to the cerebellum, as infection progressed. At least three pathophysiological components should be considered to explain this phenomenon: (i) mode of the virus spread to the target neurons in the cerebellum (i.e., Purkinje cells) and spinal cord (i.e., spinal motor neurons) from the site of intracerebral inoculation (i.e., motor thalamus) by axonal transport (i.e., anterograde and/or retrograde); (ii) ability of the virus to establish productive replication in specific neuronal cell types (i.e., PCs and SMNs), and (iii) CNS-region specificity of host responses to infection. Our

previous reconstruction of the directionality of trans-synaptic virus spread based on the neuroanatomical connectivity (*Maximova et al., 2016*) suggested that the virus may have reached the Purkinje cells by 7 dpi only by using the retrograde axonal transport (i.e., motor thalamus → deep cerebellar nuclei → Purkinje cells), while to reach the spinal motor neurons, the virus may have used the anterograde axonal transport (i.e., motor thalamus → corticospinal motor neurons → spinal motor neurons). Therefore, one possibility is that the differences in the mode and speed of axonal transport (*Black, 2016*) used by the virus to spread to target structures played a role. However, the kinetics of viral replication in the cerebellum and lumbar spinal cord were very similar (1.8 ± 0.2 and <1.7 $\log_{10}$PFU/g [at or below the limit of detection] at 3 dpi; 6.4 ± 0.3 and 5.6 ± 0.4 $\log_{10}$PFU/g at 7 dpi; and 6.9 ± 0.1 and 5.8 ± 0.2 $\log_{10}$PFU/g at 9 dpi, respectively) (*Maximova et al., 2016*; *Maximova et al., 2014*), suggesting that the virus had an equal opportunity to infect respective target neurons and to use their cellular machinery to establish productive replication. Indeed, by another measure of virus production, the amount of WNV-antigens was increasing in the Purkinje cells and spinal motor neurons at similar rates (*Maximova et al., 2016*). Thus, the possibility that these neuronal types have a different ability to support virus replication seems unlikely. Therefore, a more likely explanation of the differences in magnitude and timing of transcriptional regulation of the immune-neural-synaptic axis in WNV-infected cerebellum and spinal cord is that the host responses to infection in these CNS structures are site-specific. CNS-site-specificity in immune responses to viral infections and in mechanisms of the non-cytolytic clearance of the virus from non-renewable neurons by T and B cells has long been recognized (*Griffin, 2011*; *Griffin, 2003*; *Binder and Griffin, 2001*; *Cho et al., 2013*). Further supporting this concept, we show different patterns of T and B cell infiltration in the cerebellum versus spinal cord, as well as a differential transcriptional regulation of apoptosis of these cells.

This study may have important translational implications associated with the use of NHPs that have a high level of genetic homology to humans, parallels in functioning of the immune system, similar to humans organization of neuroanatomical pathways and skilled motor behavior and hand dexterity, all of which are coupled with their outbred nature (*Messaoudi et al., 2011*; *Lemon and Griffiths, 2005*). However, as with any animal model of disease, some limitations need to be considered before translating the findings to humans. Our NHP model of WNV-ND, in which virus is introduced directly into the neural parenchyma (*Maximova et al., 2014*), may seem artificial since in humans WNV invades the CNS and causes neurological disease following peripheral infection either through mosquito bites or after transfusion/transplantation of infected blood/organs (*Sejvar, 2016*). However, since viremia is a major neuropathogenic determinant of the flavivirus entry into the CNS (reviewed in *Maximova and Pletnev, 2018*), virus invasion of the CNS most likely occurs when immune checkpoints controlling peripheral infection and ensuing viremia (*Montgomery and Murray, 2015*; *Tobler et al., 2008*; *Qian et al., 2015*) have failed. Therefore, our intracerebral NHP model may recapitulate the pathogenesis of WNV infection in humans that takes place after virus invasion of the CNS had occurred, by bypassing the peripheral immune control mechanisms that would otherwise limit virus neuroinvasion in the vast majority of infected individuals. Indeed, less than 1% of infected people develop neuroinvasive disease after natural mosquito-borne or iatrogenic blood-borne exposure to WNV (*Sejvar, 2016*). Clearly, it is not feasible to recapitulate such exposure settings in the NHPs when 100 animals would be needed to potentially induce WNV-ND in only one animal. Nonetheless, we previously showed that the intracerebral NHP model of WNV-ND closely mimics WNV neuroinvasive disease in humans in respect to the (i) gradient in severity of affected CNS structures, (ii) neuropathology, and (iii) ensuing signs of neurological impairment (*Maximova et al., 2014*), making this model indispensable for studies of disease neuropathogenesis and testing therapeutics and vaccine safety.

Besides an immunocompromised state, age-related alterations in immune responses are the most well-defined risk factors for increased susceptibility to severe WNV-ND (*Montgomery, 2017*), but how changes associated with aging may influence regulation of the immune-neural-synaptic axis in elderly with WNV-ND warrants further investigation. Peripheral inoculation of immunodeficient or aged NHPs does not result in WNV-ND even with very high doses of WNV (*Wertheimer et al., 2010*). Therefore, future studies may address the impact of WNV infection on the immune-neural-synaptic axis in older animals using the intracerebral route of infection.

The course of WNV-ND in our animals was abruptly interrupted at the height of neurological signs (9 dpi) due to humane animal care requirements. Therefore, there is no way to know if the disease

would lead to a death or recovery, and how the phenotypic features uncovered in this study would change with time. The rapid course of WNV-ND in our NHP model is consistent with limited data showing that acute WNV encephalitis in NHPs leads to death or a moribund state that requires euthanasia 9–15 days after infection (*Arroyo et al., 2004*; *Pogodina et al., 1983*). However, a non-fatal encephalitis in NHPs, possibly associated with immunosuppression, can take a subacute course followed by virus persistence (*Pogodina et al., 1983*). It is important to underscore that even during the acute course of WNV-ND in this study, several developmental processes such as response to wounding, gliogenesis, and extracellular matrix/tissue remodeling became transcriptionally upregulated. However, at the height of neurological signs (i.e., advanced-symptomatic stage of WNV-ND) there was a downregulation rather than activation of neuronal developmental processes such as neuron development and axon guidance. Nevertheless, deeper analysis of the transcriptional regulation of established axon growth inhibitory or permissive molecules (*Anderson et al., 2016*) revealed a progressive trend which was more skewed to the axon growth permissive rather than inhibitory environment, suggesting that neuronal network repair programs may have been already initiated at the acute stage of neurological disease. Future studies should investigate correlates of recovery versus persistent neurological impairment as outcomes of WNV-ND. This may provide valuable insights and inform on how to harness potentially beneficial processes that lead to resolution of infection and neural repair.

In summary, our findings add a new dimension to understanding of regulation of the immune-neural-synaptic axis and how its homeostasis is altered during virus infection in primates. Induction of pleiotropic genes with distinct functions in each component of the immune-neural-synaptic axis suggests an unintended off-target negative impact of virus-induced immune responses on neurotransmission in the CNS. Since activation of expression of the pleiotropic genes reported here may be a part of conserved host immune responses to many viral infections, our data may serve as a resource in the search for new therapeutic approaches to restore homeostasis in interactions between the nervous and immune system at the time when virus has been cleared from the CNS.

# Materials and methods

**Key resources table**

| Reagent type (species) or resource | Designation | Source or reference | Identifiers | Additional information |
|---|---|---|---|---|
| Antibody | Mouse polyclonal anti-WNV | ATCC | Cat. #: ATCCVR-1267 AF | IHC (1:1000) |
| Antibody | Mouse monoclonal anti-synaptophysin (SY38) | Abcam | Cat. #: ab8049 RRID: AB_2198854 | IHC (1:10) |
| Antibody | Mouse monoclonal anti-MAP2 (5F9) | Millipore-Sigma | Cat. #: 05–346 RRID: AB_309685 | IHC (1:6000) |
| Antibody | Mouse monoclonal anti-CD68 (KP1) | Biocare Medical https://biocare.net/product/cd68-antibody/ | Cat. #: CM 033 | IHC (1:500) |
| Antibody | Rabbit polyclonal anti-GFAP | Agilent | Cat. #: Z0334 RRID: AB_10013382 | IHC (1:4000) |
| Antibody | Mouse monoclonal anti-CD4 (4B12) | Biocare Medical https://biocare.net/product/cd4-4b12/ | Cat. #: ACI 3148 | IHC (1:10) |
| Antibody | Rabbit polyclonal Anti-CD8 | Abcam | Cat. #: ab4055 RRID: AB_304247 | IHC (1:400) |
| Antibody | Mouse monoclonal anti-CD20 Clone L26 | Agilent | Cat. #: M0755 RRID: AB_2282030 | IHC (1:200) |
| Antibody | Rabbit polyclonal anti-Calbindin 28K | Millipore-Sigma | Cat. #: AB1778 RRID: AB_2068336 | IHC (1:1000) |
| Antibody | Rabbit monoclonal anti-ChAT Clone EPR16590 | Abcam https://www.abcam.com/choline-acetyltransferase-antibody-epr16590-ab178850.html | Cat. #: ab178850 | IHC (1:1000) |

*Continued on next page*

*Continued*

| Reagent type (species) or resource | Designation | Source or reference | Identifiers | Additional information |
|---|---|---|---|---|
| Software, algorithm | Next-Generation Clustered Heatmaps interactive tool | PMID:29092932 https://build.ngchm.net/NGCHM-web-builder/View_HeatMap.html?adv=Y | | |
| Software, algorithm | PANTHER | PMID:30804569 | RRID:SCR_004869 | |
| Software, algorithm | Reactome Knowledgebase | PMID:29145629 | RRID:SCR_003485 | |
| Software, algorithm | SynGO | PMID:31171447 | RRID:SCR_017330 | |
| Software, algorithm | gProfiler | PMID:31066453 | RRID:SCR_006809 | |

## Study design

Tissue samples from the cerebellum and spinal cord were selected from nine rhesus monkeys (*Macaca mulatta*; 2–3-year-old; seven males and two females) inoculated intrathalamically (bilaterally) with a dose of 5.0 log10 PFU of wild-type WNV strain NY99-35262 (hereafter WNV) that were used as a positive control in our prior study of the WNV vaccine safety (*Maximova et al., 2014*) and from the cerebellum and spinal cord of four rhesus monkeys (*Macaca mulatta*; 2–3-year-old; one male and three females) that were mock-inoculated intrathalamically (bilaterally) with an identical to virus inoculum volume (0.25 ml) (*Maximova et al., 2014*) of diluent without the virus (Leibovitz's L-15 medium [Invitrogen], supplemented with SPG buffer stabilizer) (detailed procedure of the bilateral intrathalamic inoculation of NHPs is described previously [*Maximova et al., 2008*]). All animal experiments were approved by the NIAID DIR Animal Care and Use Committee (animal study proposal #LID 7E). The NIAID DIR Animal Care and Use Program, as part of the NIH Intramural Research Program, complies with all applicable provisions of the Animal Welfare Act (http://www.aphis.usda.gov/animal_welfare/downloads/awa/awa.pdf) and other Federal statutes and regulations relating to animals. The NIAID DIR Animal Care and Use Program is guided by the 'U.S. Government Principles for the Utilization and Care of Vertebrate Animals Used in Testing, Research, and Training' (http://oacu.od.nih.gov/regs/USGovtPrncpl.htm).

Three WNV-infected and one mock-inoculated monkeys were euthanized at 3, 7, and 9 dpi. Detailed clinical, virological, and histopathological information about these animals can be found in our prior publications (*Maximova et al., 2014*). Tissue samples analyzed in this study were collected immediately following euthanasia and cardiac perfusion with a sterile saline in the BSL-3 environment. After removal, brains and spinal cords were aseptically dissected to be freshly preserved for RNA extraction (see later) or fixed in 10% phosphate-buffered formalin for immunohistochemistry, following the protocols similar to described previously (*Maximova et al., 2014*; *Maximova et al., 2008*). A central cerebellar coronal slice (4 mm thick) and a transverse lumbar spinal cord slice (4 mm thick) from each animal were used for RNA extraction. Immediately after dissection, each sample was placed into RNAlater (Ambion, AM7021) at 4°C. After a maximum 3 days of storage, the RNAlater was removed and tissues were stored at −80°C. For RNA extraction, the samples were thawed on ice and core tissue samples (3 mm in diameter; 4 mm thick) were extracted using sterile Harris Uni-Cores (Ted Pella, Redding, CA). The cerebellar cores were extracted from the spinocerebellar and cerebrocerebellar areas of the cerebellar cortex (including the molecular layer, Purkinje cell layer, granule cell layer, and white matter), as well as from the deep cerebellar nuclei, and pooled for each animal. The spinal cores were extracted from the ventral horns of the spinal gray matter bilaterally and pooled for each animal.

## Gene expression analysis

Microarray experiments were performed at the Research Technologies Branch, Rocky Mountain Laboratories (NIAID, NIH). The miRNeasy Mini kit (Qiagen) was used to extract total RNA via the QIAcube robot (Qiagen). To prepare target, 50 ng of each RNA sample was used as template for the Ovation V2 RNA Amplification System (Nugen, Cat#3100) to make amplified cDNA, which was purified using QIAquick 96-well (Qiagen) protocol. The cDNA (3.75 µg) was fragmented and labeled using the Encore Biotin Labeling Kit (Nugen, Cat# 4200), hybridized onto the Rhesus Macaque

GeneChip (Affymetrix, P/N 90065), washed, and scanned according to manufacturer's instructions. Microarray data were normalized using Affymetrix Expression Console Software and gene expression analyzed using Affymetrix Transcriptome Analysis Console (Santa Clara, CA). Differentially expressed transcripts identified by ANOVA were arithmetically averaged and compared ratiometrically to average expression in control tissue for visualization using Spotfire Analyst (TIBCO; Palo Alto, CA) and fold changes (FC) $\leq-2$ and $\geq2.0$, false discovery rate (FDR) < 0.05 were used as cutoffs to define the significantly DEGs for subsequent functional genomic analyses. The NHP gene expression data have been deposited in NCBI's Gene Expression Omnibus (*Edgar et al., 2002*) and are accessible through GEO Series accession number GSE122798 (https://www.ncbi.nlm.nih.gov/geo/query/acc.cgi?acc=GSE122798).

## Genomic analyses

Genomic heat maps were created using the Next-Generation Clustered Heatmaps interactive tool (*Broom et al., 2017*) https://build.ngchm.net/NGCHM-web-builder/View_HeatMap.html?adv=Y. The Reactome Knowledgebase (*Fabregat et al., 2018*) was used to identify and visualize the significantly affected biological domains and pathways (p<0.05). The PANTHER classification system (*Mi et al., 2019*) was used to identify significantly enriched (FDR < 0.05) GO terms by the ORT and the statistical enrichment test based on the Mann–Whitney U-test (Wilcoxon Rank-Sum test). The SynGO knowledgebase for the synapse (*Koopmans et al., 2019*) was used to determine the enrichment (FDR < 0.05) and to identify, annotate, and analyze the structural locations and functions of the significantly differentially expressed synaptic genes. As a background reference list, we used a list of genes expressed in the brain. This brain-expressed gene list contains 16,036 genes and is defined as 'expressed in any GTEx v7 brain tissue'. This list was generated by ranking gene-expression levels in the brain versus other tissues (*Ganna et al., 2016*) using Genotype-Tissue Expression Consortia (GTEx) data (*Lonsdale et al., 2013*), and was kindly provided by Dr. Frank Koopmans (VU University and UMC Amsterdam). gProfiler (*Raudvere et al., 2019*) was used to identify the GO terms and pathways that were significantly enriched (FDR < 0.05) for the cell death processes and for the up- or downregulated pleiotropic immune-neural-synaptic DEGs.

## Immunohistochemistry

Following euthanasia and cardiac perfusion with sterile saline, cerebellar and spinal tissue samples were immediately collected, fixed on 10% formalin for 7 days and processed for immunohistochemistry. Brightfield immunohistochemistry was performed following previously described procedures (*Maximova et al., 2008*). The following primary antibodies were used: WNV-specific primary antibodies in hyperimmune mouse ascitic fluid (ATCCVR-1267 AF; 1:1000); anti-synaptophysin (mouse monoclonal [SY38]; Abcam; 1:10); anti-MAP2 (mouse monoclonal [5F9]; Millipore-Sigma; 1:6000); anti-CD68 (mouse monoclonal [KP1]; Biocare Medical; 1:500); anti-GFAP (rabbit polyclonal; Agilent; 1:4000); anti-CD4 (mouse monoclonal; Biocare Medical; 1:10); anti-CD8 (rabbit polyclonal; Abcam; 1:400); and anti-CD20 (mouse monoclonal; Agilent; 1:200). Diaminobenzidine was used for colorimetric detection (brown) of each protein marker. Sections were counterstained with hematoxylin. Whole tissue section imaging was performed at ×40 magnification using the ScanScope AT2 (Leica Biosystems). Aperio eSlide Manager and ImageScope software were used for digital slide organization, viewing, acquisition, and analysis. Double immunofluorescent staining to identify WNV-infected neuronal cell types was performed using Bond RX (Leica Biosystems) according to manufacturer protocols and with the following primary antibodies: WNV-specific primary antibodies in hyperimmune mouse ascitic fluid (ATCCVR-1267 AF; 1:1000) (for WNV-antigens) and Calbindin 28K (rabbit polyclonal; Millipore-Sigma; 1:1000) (for Purkinje cells) or ChAT (rabbit monoclonal; Abcam [clone EPR16590]; 1:1000) (for spinal motor neurons) and host appropriate secondary antibodies labeled with red fluorescent dye Alexa Flour 594 (Life Technologies; 1:300) or biotinylated secondary antibody (Vector Laboratories; 1:200) and green fluorochrome streptavidin 488 (Life Technologies; 1:500) and DAPI nuclear counterstain (Vector Laboratories).

## Validation of microarrays

Template cDNAs were synthesized from RNA samples using SuperScript VILOTM cDNA synthesis kit (ThermoScientific, Waltham, MA). Resulting cDNAs were purified using the QIAquick 96 PCR

Purification Kit according to the manufacturer's protocol (Qiagen, Valencia, CA). These purified cDNAs were used in the TaqMan qPCR validation assay. cDNAs were analyzed by qPCR using the reference gene TAF11 primers and probe in duplex with primers and probe for each select gene. Linear gene-to-reference gene ratios were calculated for each gene and sample. Normalized microarray values were used to calculate gene-to-reference ratios for each sample and gene. Spearman correlation (GraphPad Software, La Jolla, CA) was calculated between qPCR and microarray values. The list of primers is provided in *Supplementary file 10* ('WNV-ND transcriptome validation').

## Acknowledgements

We thank Dzung Thach, Stephen Porcella, and Timothy Myers for initial sample preparation and technical support; and Dr. Shari Price-Schiavi and the staff of the Pathology Associates Division of Charles River Laboratories (Frederick, MD) for technical support with brightfield immunohistochemistry. This work was supported by the NIAID Intramural Research Program.

## Additional information

### Funding

| Funder | Grant reference number | Author |
| --- | --- | --- |
| National Institute of Allergy and Infectious Diseases | Intramural Research Program | Olga A Maximova<br>Daniel E Sturdevant<br>John C Kash<br>Kishore Kanakabandi<br>Yongli Xiao<br>Mahnaz Minai<br>Ian N Moore<br>Jeff Taubenberger<br>Craig Martens<br>Jeffrey I Cohen<br>Alexander G Pletnev |

The funders had no role in study design, data collection and interpretation, or the decision to submit the work for publication.

### Author contributions

Olga A Maximova, Conceptualization, Resources, Data curation, Software, Formal analysis, Supervision, Funding acquisition, Validation, Investigation, Visualization, Methodology, Writing - original draft, Project administration, Writing - review and editing; Daniel E Sturdevant, Data curation, Formal analysis, Validation, Visualization, Methodology, Writing - review and editing; John C Kash, Data curation, Formal analysis, Visualization, Methodology; Kishore Kanakabandi, Data curation, Formal analysis, Validation, Investigation; Yongli Xiao, Data curation; Mahnaz Minai, Methodology, Performed double immunofluorescent staining; Ian N Moore, Supervision, Oversaw double immunofluorescent staining; Jeff Taubenberger, Resources, Supervision; Craig Martens, Resources, Supervision, Validation, Project administration, Writing - review and editing; Jeffrey I Cohen, Alexander G Pletnev, Resources, Supervision, Funding acquisition, Writing - review and editing

### Author ORCIDs

Olga A Maximova (iD) https://orcid.org/0000-0002-8906-9738

### Ethics

Animal experimentation: All animal experiments were approved by the NIAID DIR Animal Care and Use Committee (animal study proposal #LID 7E). The NIAID DIR Animal Care and Use Program, as part of the NIH Intramural Research Program, complies with all applicable provisions of the Animal Welfare Act (http://www.aphis.usda.gov/animal_welfare/downloads/awa/awa.pdf) and other Federal statutes and regulations relating to animals. The NIAID DIR Animal Care and Use Program is guided by the "U.S. Government Principles for the Utilization and Care of Vertebrate Animals Used in Testing, Research, and Training" (http://oacu.od.nih.gov/regs/USGovtPrncpl.htm).

Decision letter and Author response
Decision letter https://doi.org/10.7554/eLife.62273.sa1
Author response https://doi.org/10.7554/eLife.62273.sa2

## Additional files

### Supplementary files
- Supplementary file 1. Coordinated transcriptional shifts.
- Supplementary file 2. Synaptic genes differentially expressed during WNV-ND.
- Supplementary file 3. SynGO enrichment analysis results.
- Supplementary file 4. SynGO enrichment: up- and downregulated synaptic genes.
- Supplementary file 5. Venn-diagram results for immune and neural DEGs.
- Supplementary file 6. ORT: Immune-neural pleiotropic DEGs.
- Supplementary file 7. Immune-neural-synaptic pleiotropic genes.
- Supplementary file 8. gProfiler: Immune-neural-synaptic pleiotropic DEGs.
- Supplementary file 9. Pleiotropic DEGs: Synaptic topology and functionality.
- Supplementary file 10. WNV-ND transcriptome validation.
- Transparent reporting form

### Data availability
The NHP gene expression data have been deposited in NCBI's Gene Expression Omnibus and are accessible through GEO Series accession number GSE122798 (https://www.ncbi.nlm.nih.gov/geo/query/acc.cgi?acc=GSE122798 ).

The following dataset was generated:

| Author(s) | Year | Dataset title | Dataset URL | Database and Identifier |
|---|---|---|---|---|
| Maximova OA, Sturdevant DE, Kash JC, Kanakabandi K, Xiao Y, Minai M, Moore IN, Taubenberger J, Martens C, Cohen JI, Pletnev AG | 2021 | Virus infection of the CNS disrupts the immune-neural-synaptic axis via induction of pleiotropic gene regulation of host responses | https://www.ncbi.nlm.nih.gov/geo/query/acc.cgi?acc=GSE122798 | NCBI Gene Expression Omnibus, GSE122798 |

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
