## [Decision Letter]

**Acceptance summary:**

Your studies of West Nile Virus infection in the primate brain provide novel insights into the devastating consequences of this disease in particular and perhaps that of other viral infections more broadly. The detailed analysis of gene expression changes at multiple time points after infection reveals a complex network of responses and will provide valuable information for other researchers pursuing additional modes of analysis.

**Decision letter after peer review:**

Thank you for submitting your article "Virus infection disrupts the immune-neural-synaptic axis via induction of pleiotropic gene regulation of host responses" for consideration by *eLife*. Your article has been reviewed by three peer reviewers, one of whom is a member of our Board of Reviewing Editors, and the evaluation has been overseen by Tadatsugu Taniguchi as the Senior Editor. The following individual involved in review of your submission has agreed to reveal their identity: Alan Barrett (Reviewer #2).

The reviewers have discussed the reviews with one another and the Reviewing Editor has drafted this decision to help you prepare a revised submission.

Summary:

Maximova et al. report on studies using a non-human primate model of encephalitis due to West Nile virus (WNV). WNV is the leading cause of arboviral neuroinvasive disease in the US with post-infectious neurologic sequelae including movement disorders and memory impairments. The authors performed intracerebral inoculation of a virulent strain of WNV and performed extensive analyses of WNV-induced alterations in gene expression on cerebellar and spinal cord tissues at 3, 7, and 9 days post-infection (dpi) compared with mock-infected animals. They also performed immunohistochemical studies examining astrocytes, microglia and T cells and neuronal infection, loss and synapse elimination.

The paper is very well-written, supports findings in murine models and human disease, which have shown that presynaptic termini are eliminated in the WNV-infected hippocampus. That these processes may occur in multiple CNS regions is clinically relevant and supports the need for translation of these findings into new targets that might address multiple neurologic symptoms and sequelae.

This is a largely descriptive study on the transcriptional response of two brain regions, the cerebellum and spinal cord, to i.c.v. infusion of West Nile virus. The strength of the study is the use of non-human primates, analysis at two time-points post infection and the importance of the topic in light of the current global viral pandemic. Analytics that focus on particular cellular components, thereby identifying genes associated with pre- and post-synaptic compartments as particularly strongly regulated targets, is another strength. The use of IHC to confirm predicted changes in protein levels under girding astrocytosis, microglia and T-cell reactivity and cell death is a further strength.

Essential revisions:

1) It is important to remember that WNV only causes disease in NHPs when the virus is directly inoculated into the brain (as pointed out by the authors). This raises important questions about the contribution of putting a needle directly in the brain and how this affects gene regulation. Unfortunately, the authors do not tell us if the control animals also received a needle injection into the brain too (nor does Maximova et al., 2014, the original study). This is an important point to consider in terms of the validity of the data. It is interesting that changes were detected in the spinal cord before the cerebellum in virus infected animals plus no gene changes in the cerebellum at day 3pi supporting the possibility that the needle injection does not contribute to the data obtained. However, this needs to be formally addressed in the revision.

2) Related to the above, there is limited information as to how well the i.c.v. infusion of virus replicates WNV-ND in humans and this too should be more formally addressed. The Discussion provides a justification for the system being used. The reviewers note this is an artificial system. Humans get WN disease following a mosquito bite and the virus crosses the periphery into the brain. Humans do not get the virus infection by direct injection into the brain. I think this paragraph needs some revisions.

3) The manuscript would benefit from more cellular analyses to localize synaptic proteins within microglia or astrocytes within the CB and SC, and their relationship with clinical findings. Presumably the authors have existing tissue which could be further analyzed to address, at least in part, some of the following issues:

a) Figure 4. The authors show extensive astrocyte and microglial activation in both CNS regions of infected NHPs. Given the microglial nodules observed, can they also determine whether there was any observed neuronophagia in these tissues?

b) Figure 5. The loss of synaptophysin is very striking. Can the authors also look at post-synaptic markers to determine whether both sides of the synapse are affected? It would also be important to determine whether synaptic elements can be detected within microglia, as has been observed in murine models of WNV CNS infection.

c) Figure 6. The data demonstrating differential regulation of pre- and post-synaptic genes is very striking. As mRNAs for synaptic genes are localized in neurites, it would be important to determine whether the pre-synaptic termini are eliminated more efficiently and the post-synaptic termini spared.

d) Figure 7. As a follow-up, it would be important to examine locations of synapse elimination within both CNS regions and relate this to motor findings in the animals.

Also, given the role of IL-1 in neurotoxicity (shown in both in vitro and WNV models; 10.1038/nature21029 and 10.1038/s41590-017-0021-y), it would be interesting to determine cellular sources of this cytokine in the NHP model.

4) Figure 1. The authors mention that animals are euthanized when they develop severe neurologic symptoms. Please comment on whether the model would lead to universal death in NHPs if this were not performed. This is important because it is unclear whether patient survivors exhibit the same loss of neurons observed in those who succumb during acute CNS infection. Can the authors report on whether expression of the same genes is altered in both CNS regions during symptomatic stages of disease? Also, as many genes designated as “immune systems” function in the CNS during development and repair, it would be important to categorize them as “ׅneuroimmune systems”, which is consistent with increasing evidence of ancestral functions of neural genes as immune genes. It would also be helpful to highlight those known to function during development as they may be involved in repair.

5) Figure 2. This figure shows upregulated immune system genes and down-regulated nervous system genes. It would be important to look at both up and down in both systems for the two regions. It is interesting that the CB seems to lag behind the spinal cord with regard to innate immune responses such as activation of TLRs. Can the authors comment on the time-frame of appearance of symptoms in the animals? That is, do spinal cord deficits precede cerebellar deficits? It also appears that cytokine signaling is quite different between the two regions. Studies in mice suggest that the CB is far less targeted by WNV and has constitutive expression of many PRRs. Is this the case in NHPs?

6) Figure 3. The results of these analyses are consistent with published studies showing persistent activation of innate and adaptive immune responses after viral clearance. The more subtle changes in nervous system genes are also consistent with results suggesting that survivors of WNV may not have extensive neural cell damage but instead exhibit alterations in neuronal networks. Can the authors provide any additional analyses to evaluate this such as examining cell death pathways in neurons?

7) That the genes which are being regulated have multiple functions, i.e. are pleiotropic, does not seem particularly surprising. The authors argue this leads to unintended deleterious effects but the logical connection there is not clear, why could it not be the opposite? Actually providing a benefit?

---

## [Author Response]

Essential revisions:1) It is important to remember that WNV only causes disease in NHPs when the virus is directly inoculated into the brain (as pointed out by the authors). This raises important questions about the contribution of putting a needle directly in the brain and how this affects gene regulation. Unfortunately, the authors do not tell us if the control animals also received a needle injection into the brain too (nor does Maximova et al., 2014, the original study). This is an important point to consider in terms of the validity of the data. It is interesting that changes were detected in the spinal cord before the cerebellum in virus infected animals plus no gene changes in the cerebellum at day 3pi supporting the possibility that the needle injection does not contribute to the data obtained. However, this needs to be formally addressed in the revision.

The control animals used for this study were mock-infected (not an un-operated normal control). That is, the procedure for intracerebral (i.e., bilateral intrathalamic) was identical (i.e., needle insertion and injection of the inoculum volume) for all animals used in the study. The only difference between the virus-infected and mock-infected animals was that the inoculum injected into the thalami of the mock-control animals contained only a diluent but not the virus (1) (detailed procedure is described by us previously (2)). Thus, a potential artifact of needle insertion and injection of the inoculum volume can be excluded since we designed the study to be well-controlled and all analyses are based on the comparisons of the virus-infected animals and mock-infected animals (i.e., gene expression in virus-infected CNS upregulated or downregulated *over* mock-infected and protein expression in virus-infected CNS compared to mock-infected at the same time point after inoculation).

We added a relevant information (and the reference describing our procedure of intracerebral inoculation of NHPs in details) to formally address and clarify this in the Results and Materials and methods sections:

Results:

“The mock control cerebellum or spinal cord samples were from NHPs (n=4) that were inoculated intracerebrally in an identical manner to the virus-inoculated animals, except that the inoculum contained only diluent and not virus (1) (detailed procedure is described by us previously (2)). These four mock-inoculated control animals were used for normalization of gene expression and one animal (euthanized at 10 dpi) was used as a normal control for immunohistochemistry that examined the advanced-symptomatic stage of WNV-ND (9 dpi)”.

Materials and methods:

“Tissue samples from the cerebellum and spinal cord were selected from nine rhesus monkeys (*Macaca mulatta*; 2-3-year-old; 7 males and 2 females) inoculated intrathalamically (bilaterally) with a dose of 5.0 log10 PFU of wild-type WNV strain NY99-35262 (hereafter WNV) that were used as a positive control in our prior study of the WNV vaccine safety (1) and from the cerebellum and spinal cord of four rhesus monkeys (*Macaca mulatta*; 2-3-year-old; 1 male and 3 females) that were mock-inoculated intrathalamically (bilaterally) with an identical to virus inoculum volume (0.25 ml) (1) of diluent without the virus (Leibovitz’s L-15 medium [Invitrogen], supplemented with SPG buffer stabilizer) (detailed procedure of the bilateral intrathalamic inoculation of NHPs is described previously (2)).”

2) Related to the above, there is limited information as to how well the i.c.v. infusion of virus replicates WNV-ND in humans and this too should be more formally addressed. The Discussion provides a justification for the system being used. The reviewers note this is an artificial system. Humans get WN disease following a mosquito bite and the virus crosses the periphery into the brain. Humans do not get the virus infection by direct injection into the brain. I think this paragraph needs some revisions.

To address these comments, we removed lines from the Discussion and revised the paragraphs where we discuss the limitations of our animal model, as follows:

“This study may have important translational implications associated with the use of nonhuman primates that have a high level of genetic homology to humans, parallels in functioning of the immune system, similar to humans organization of neuroanatomical pathways and skilled motor behavior and hand dexterity, all of which are coupled with their outbred nature (3, 4). […] Therefore, future studies may address the impact of WNV infection on the immune-neural-synaptic axis in older animals using the intracerebral route of infection”.

3) The manuscript would benefit from more cellular analyses to localize synaptic proteins within microglia or astrocytes within the CB and SC, and their relationship with clinical findings. Presumably the authors have existing tissue which could be further analyzed to address, at least in part, some of the following issues:

We strongly agree that our understanding of the roles of microglia and astrocytes in the pathogenesis of WNV infection of the CNS would benefit from further studies. However, we believe that the suggestion to address the outstanding issues listed below would require extensive analyses which should encompass methodologies that are more comprehensive that suggested by reviewers in order to get meaningful insights. Main focus of our current study was to reveal the transcriptional dysregulation of the CNS homeostasis and alterations in regulation of neural functions when the viral infection of neurons is introduced into the equation. Clearly, as we believe could be seen below from our responses, the investigation of the mechanisms of synaptic loss/elimination and identification of major cell players involved would be not a trivial task and is well beyond the scope of this study.

a) Figure 4. The authors show extensive astrocyte and microglial activation in both CNS regions of infected NHPs. Given the microglial nodules observed, can they also determine whether there was any observed neuronophagia in these tissues?

In our opinion, neuronophagia is an old and not very meaningful term since it can only describe the dying neurons surrounded by other cells, presumably phagocytic, in the routinely histologically stained (i.e., H&E or Nissl) sections. We do not describe the microglial nodules (another somewhat outdated and misleading term, in our opinion) but went further in analyzing the topographical relation of the activated and phagocytic microglial cells to infected neurons and presented a detailed results in Figure 6B and Table 1, showing that at the advanced-symptomatic stage of WNV-ND, activated microglia migrated toward infected Purkinje and spinal motor neurons (neuron-centripetal migration) and assumed the perineuronal topology. However, a close apposition of the activated microglia to neurons does not necessarily mean that the sole action of these cells is to phagocytose the neurons and/or their synapses. In fact, ascribing such roles to microglia can be a case of guilt by association (5) until proven otherwise. We acknowledge that more studies are needed to decipher the roles of microglia in various viral infections of the CNS, but stress that the goal of current study was to find out whether the microglia were activated at the protein level (CD68 immunostaining) and whether their cell morphology, migration pattern, and topology corresponded to the functional states revealed by changes in gene expression.

b) Figure 5. The loss of synaptophysin is very striking. Can the authors also look at post-synaptic markers to determine whether both sides of the synapse are affected? It would also be important to determine whether synaptic elements can be detected within microglia, as has been observed in murine models of WNV CNS infection.c) Figure 6. The data demonstrating differential regulation of pre- and post-synaptic genes is very striking. As mRNAs for synaptic genes are localized in neurites, it would be important to determine whether the pre-synaptic termini are eliminated more efficiently and the post-synaptic termini spared.

We also found the pattern of loss of synaptophysin immunoreactivity in relation to the topography of WNV-infected neurons very striking. We elected to probe for the synaptophysin in order to validate our most striking finding showing that the synapse was the most transcriptionally dysregulated cellular compartment in neurons during acute symptomatic WNV infection (which may include either presynaptic axonal terminals originating from virus infected neurons or such terminals from uninflected neurons that innervate infected neurons, or both). Advantages of using this protein marker to confirm synaptic dysfunction at the gene expression level in this study were twofold: (i) synaptophysin is an integral component of synaptic vesicles and the most established protein marker used to assess the synaptic integrity; and (ii) synaptophysin is a pan-synaptic protein which is present in virtually all types of synapses (thus providing a full coverage of synaptic types present in two distinct neural networks studied (i.e., cerebellar circuitry which provides the control of movement and spinal motor neuron circuitry which constitutes a final path in the execution of movement). In addition, during the revision, we added the results of immunostaining for the choline acetyltransferase which, in addition to being an established protein marker for the spinal motor neurons, also provided a means to assess the integrity of one of the largest and numerous synaptic terminals innervating these neurons – cholinergic C-boutons. Together, the dramatic decrease of immunoreactivity for these proteins provide a strong support to the extensive transcriptional dysregulation of synaptic compartments revealed in this study.

However, given the enormous number of currently known synaptic proteins (1,112 according to the most comprehensive synaptic gene ontology database that we used in this study (6)), straining for any single presynaptic molecule (number of currently known presynaptic genes: 482) or postsynaptic molecule (number of currently known presynaptic genes: 592) would neither give a full picture of synaptic dysregulation, nor it would answer the question whether the pre- or postsynaptic compartments are preferentially eliminated or spared and by which mechanisms.

In our opinion, addressing the issue of presynaptic terminals and/or their postsynaptic partners loss or retention, as well as synaptic elimination by microglia (and/or astrocytes) by immunostaining for a single pre- or postsynaptic protein (e.g., in the hippocampus of mice infected with a modified WNV (7)) and conventional fluorescent microscopy does not provide a sufficient resolution to draw definitive conclusions. Not only the loss of a single protein expression from each synaptic compartment should not be judged as a loss of entire compartment (see our argument above for the enormous numbers of currently established proteins in each compartment), presence of microglia in close proximity to the synapses should not be judged as evidence of active stripping of synapses by microglial cells (5). Indeed, a high-resolution correlative light and electron microscopy have already challenged such conclusions (8)). Therefore, we believe the above issues should be better addressed by the highest resolution methods such as electron microscopy at the morphological level and a high throughput proteomics at a functional level. Notably, previously published attempts with using the electron microscopy (7) does not, in our opinion, provide a convincing evidence of synapse elimination by microglia, but rather show microglial cells and their processes only “adjacent” or “surrounding” the synapse, again, consistent with a case of a guilt by association (5).

On the other hand, presence of synaptic proteins within the microglia may reflect a secondary effort by these cells that acquired an activated phagocytosis phenotype (confirmed in this study by detecting upregulation of phagocytosis at gene expression level and high expression of phagocytosis protein CD68) to clean up already dysfunctional and detached synapses and thus should be more appropriately viewed as a consequence rather than a cause. In line with this, we identified a dysregulation of protein-protein interactions at synapse and downregulation of synapse adhesion molecules (neurexins and neuroligins) between pre- and post-synapse in this study (Figure 3 (nodes #8 and #10 in the “Neuronal system domain”); Tables 2 and 3), suggesting the detachment of the presynaptic terminals from their postsynaptic partners as possible mechanism of synaptic dysfunction and disruption of the neurotransmission. Whether the neurons orchestrate their own synapse elimination (9) by glial cells (i.e., microglia and astrocytes), infiltrated cytotoxic lymphocytes, or any combination of these cells in specific developmental or pathological conditions (7, 9-17), or such elimination represents a secondary clean up events after the presynaptic terminal were already detached from their postsynaptic partners, is an open question.

Clearly, addressing all these outstanding questions is beyond of the scope of our current manuscript, but we enthusiastically agree with reviewers’ comments that such insights are highly desired and should be a focus of the dedicated follow up studies. Adding to a complexity of such future research, the virus infection induced changes must be studied in the context of each specific neural circuitry since their synaptic type composition and respective functions will be unique.

d) Figure 7. As a follow-up, it would be important to examine locations of synapse elimination within both CNS regions and relate this to motor findings in the animals.

We agree that this will have to be deciphered in the future follow-up studies, but current staining patterns for the synaptophysin and choline acetyltransferase suggest the loss of synaptic connections that innervate specific neuronal types infected by WNV – Purkinje cells in the cerebellar cortex and spinal motor neurons in the gray matter of ventral horns of the spinal cord. We added the lists of putatively affected synapses innervating the Purkinje cells and spinal motor neurons in Table 1 and following text:

“Topography of the observed loss of the presynaptic compartments (Figure 7D; Figure 8A and C) and postsynaptic cellular targets (Figure 8B and D) in WNV-ND suggests that (i) putatively affected synapses in the cerebellar cortex may include synapses innervating Purkinje cells (e.g., climbing fiber-PC and parallel fiber-PC [glutamatergic, asymmetric, axo-dendritic, and excitatory] in the molecular layer, Basket cell-PC [GABA-ergic and inhibitory] in the molecular and PC layer, and stellate cell-PC [GABA-ergic and inhibitory] in the molecular layer; and (ii) putatively affected synapses in the spinal cord may include synapses innervating the spinal motor neurons (e.g., cholinergic C-boutons, glutamatergic synapses from descending tracts, and inhibitory synapses from local inhibitory neuron networks) (Table 1).”

Also, given the role of IL-1 in neurotoxicity (shown in both in vitro and WNV models; 10.1038/nature21029 and 10.1038/s41590-017-0021-y), it would be interesting to determine cellular sources of this cytokine in the NHP model.

In this work, we did not cherry-pick the molecules to analyze. However, our transcriptome data will provide an extensible resource for future data mining and validation of cell sources and the data will be publicly available at the time of publication.

4) Figure 1. The authors mention that animals are euthanized when they develop severe neurologic symptoms. Please comment on whether the model would lead to universal death in NHPs if this were not performed. This is important because it is unclear whether patient survivors exhibit the same loss of neurons observed in those who succumb during acute CNS infection.

We commented on this point in the original Discussion:

“The course of WNV-ND in our animals was abruptly interrupted at the height of neurological signs (9 days postinfection) due to humane animal care requirements. […] However, a non-fatal encephalitis in NHPs, possibly associated with immunosuppression, can take a subacute course followed by virus persistence (18)”.

Can the authors report on whether expression of the same genes is altered in both CNS regions during symptomatic stages of disease?

We added the Venn diagram comparisons of the genes that were upregulated or downregulated in the cerebellum and spinal cord during symptomatic stages of disease (Figure 1F and H). This analysis showed that:

“Similarity between the genes that were differentially expressed in the WNV-infected cerebellum and spinal cord was higher for the upregulated genes (40 – 42%), compared to overlaps in the downregulated genes (9 – 12%)”.

Also, as many genes designated as “immune systems” function in the CNS during development and repair, it would be important to categorize them as “neuroimmune systems”, which is consistent with increasing evidence of ancestral functions of neural genes as immune genes.

As mentioned in the discussion, our main goal was to “to deconvolute the complex changes in CNS physiology that occur during flavivirus infection by examining differential regulation of biological processes related to the immune and nervous systems”. To this end, we relied on the categorization of differentially expressed genes established by their respective immune or neural gene ontologies. To further clarify this, we added the following to conclusions in the Results section:

“Together, these results show that WNV infection induces progressive and CNS-structure-dependent gene expression changes that manifest in transcriptional upregulation of biological pathways ascribed to the immune system gene ontologies and downregulation of functions related to the neuronal system biological domain.”

It would also be helpful to highlight those known to function during development as they may be involved in repair.

To address this suggestion, we performed additional study and added new results and discussion:

Results:

“Since the top largest enriched GO BP term for downregulated genes was “nervous system development“, we further dissected transcriptional regulation of the developmental and repair processes in the CNS that may be altered by WNV infection using the gProfiler (19). […] Taking into account the fold changes for these 22 axon growth genes, when (i) downregulation of permissive or upregulation of inhibitory molecules would indicate the inhibitory molecular environment for axonal regeneration, while (ii) downregulation of inhibitory or upregulation of permissive molecules would indicate the permissive environment, we found a progressive trend which was skewed to the axon growth permissive environment.”

Results:

“In addition, we show that WNV infection alters transcriptional regulation of developmental and repair processes in the CNS, which are associated with upregulation of the immune system development/wound healing and downregulation of the neuronal/axonal regenerative processes. However, further dissection of the molecular environment associated with axonal regeneration also indicated a trend that was more skewed to the axon growth permissive rather than inhibitory environment, suggesting initiation of the neuronal network repair programs.”

Discussion:

“It is important to underscore that even during the acute course of WNV-ND in this study, several developmental processes such as response to wounding, gliogenesis, and extracellular matrix/tissue remodeling became transcriptionally upregulated. […] This may provide valuable insights and inform on how to harness potentially beneficial processes that lead to resolution of infection and neural repair. “

5) Figure 2. This figure shows upregulated immune system genes and down-regulated nervous system genes. It would be important to look at both up and down in both systems for the two regions.

Both upregulated and downregulated genes for the two CNS regions (i.e., cerebellum and spinal cord) at both the early- and advanced-symptomatic stages of WNV-ND were analyzed for enrichment using the Reactome Knowledgebase. Original Figure 2 presented only statistically significant enrichment for both the immune and neuronal systems. We clarified this by adding two sentences:

Sentence #1: “As expected based on the identification of the immune system as the top biological system significantly enriched for the upregulated genes (Figure 1G), Reactome pathways associated with this system showed significant enrichment only for upregulated genes (Figure 3C, D, G and H), but not for the downregulated genes (at both early-symptomatic and advanced-symptomatic stages of WNV-ND, and in both the cerebellum and spinal cord).”

Sentence #2: “As expected based on the identification of the neuronal system as the top biological system significantly enriched for the downregulated genes (Figure 1H), Reactome pathways associated with this system showed significant enrichment only for downregulated genes (Figure 3E, F, I and J), but not for upregulated genes (at both early-symptomatic and advanced-symptomatic stages of WNV-ND, and in both the cerebellum and spinal cord).”

It is interesting that the CB seems to lag behind the spinal cord with regard to innate immune responses such as activation of TLRs. Can the authors comment on the time-frame of appearance of symptoms in the animals? That is, do spinal cord deficits precede cerebellar deficits?

The clinical time course in rhesus monkeys after bilateral intrathalamic inoculation of 5 log PFU of the wild type WNV was very rapid, with (i) onset of the clumsiness and lethargy on day 4 postinoculation, (ii) shaky movements, incoordination, and limb weakness on day 6 postinoculation; and (iii) worsening of the above neurological signs leading to a moribund state by day 9-10 postinoculation (1). Given such a rapid development of neurological deficits, constrains of the level-3 biological containment, and the fact that it is unpractical to administer a comprehensive neurological exam to rhesus monkeys to test for specific cerebellar and spinal cord deficits, we cannot judge whether the deficits associated with the cerebellar or spinal neural networks preceded one another. The observed neurological signs such clumsiness, shaky movements, incoordination, and limb weakness can be attributed to the deficits in the cerebellar and/or spinal functions. Interestingly, we previously showed that WNV spread from the thalamus (site of inoculation) occurs most probably transsynaptically by both anterograde and retrograde axonal transport (20). Both the cerebellar neurons (specifically, Purkinje cells) and spinal motor neurons represent the second order of neurons reached by WNV (can be seen in Figure 8A in reference (20)). However, virus spread to the Purkinje cells in the cerebellum could occur only by the retrograde axonal transport, while virus most likely used the anterograde axonal transport to spread to the spinal cord motor neurons. Adding to a complexity, the speed of these modes of axonal transport and the length of neuroanatomical pathways used by the virus to reach the cerebellum and spinal cord (especially the lumbar region studied here) should be considered. Nevertheless, the kinetics of viral replication in the cerebellum and lumbar spinal cord was similar and virus titers reached were among the highest compared to all other CNS regions studied (can be seen in Figure 2 in reference (1)). Thus, the cerebellar lagging behind the spinal cord cannot be simply explained and awaits further elucidation.

It also appears that cytokine signaling is quite different between the two regions. Studies in mice suggest that the CB is far less targeted by WNV and has constitutive expression of many PRRs. Is this the case in NHPs?

The “Cytokine signaling in Immune System” node (node #3 in Figure 3) became progressively significantly enriched (that is, more specific pathway levels became significantly enriched) over time course of symptomatic WNV-ND in both the cerebellum and spinal cord. This was also the true for the ”child” nodes including the “Interferon Signaling” (node #4 in Figure 3) and “Signaling by Interleukins” (node #5 in Figure 3). Pathways in the “Interferon Signaling” nodes (including the “Antiviral mechanism by IFN-stimulated genes”, “Interferon alpha/beta signaling”, and “Interferon gamma signaling”) were also enriched in both cerebellum and spinal cord during symptomatic WNV-ND. Slightly less enrichment in specific interleukin family pathways in the cerebellum most likely does not constitute a crucial difference in the cytokine signaling between these two CNS regions.

Since no specifics provided in referring to the studies in mice that suggest that the cerebellum is less targeted by WNV in mouse models, it is difficult to comment on that. However, we aware that the granule cell neurons in the cerebellum (not the Purkinje cells, which are targeted by WNV and efficiently support its replication) have been studied ex vivo and in vivo in mice and shown to have unique innate immune signatures (high expression and epigenetic regulation of the interferon-stimulated genes), that correlated with enhanced antiviral response in this type of the cerebellar neurons and rendered them less permissive to infection (21). This underscores the fact that the cerebellum is not “far less targeted by WNV”, and that the Purkinje cells but not granule cell neurons are major targets of WNV in the cerebellum in NHPs (20) and humans (22) and that Purkinje cells but not granule cell neurons are responsible for very high virus loads in the cerebellum of NHPs (1, 20).

6) Figure 3. The results of these analyses are consistent with published studies showing persistent activation of innate and adaptive immune responses after viral clearance. The more subtle changes in nervous system genes are also consistent with results suggesting that survivors of WNV may not have extensive neural cell damage but instead exhibit alterations in neuronal networks. Can the authors provide any additional analyses to evaluate this such as examining cell death pathways in neurons?

We performed additional analyses to examine transcriptional regulation of the cell death in the cerebellum and spinal cord during the advanced-symptomatic stage of WNV-ND with a particular focus on neurons.

The results are described in the new section: “Cell death processes in WNV-ND are regulated bi-directionally and magnitude of neuron cell death regulation does not exceed that of lymphocytes”.

In summary, these results demonstrated that “activation of transcriptional regulation of the cell death in WNV-ND is (i) bi-directional (i.e., concurrently positive and negative), (ii) region-specific (i.e., skewed to a positive regulation in the cerebellum but negative regulation in the spinal cord), (iii) cell-type-specific (i.e., major cell types with increased regulation of the cell death processes were neurons and lymphocytes). […] Strikingly, regulation of the apoptotic processes in lymphocytes was a region-specific, with increased regulation of T-cell apoptosis in the cerebellum but not in the spinal cord and vice versa – increased regulation of B-cell apoptosis in the spinal cord but not in the cerebellum, suggesting differential lymphocytic responses to WNV infection in these two CNS regions.”

Thus, a closer examination of transcriptional regulation of the cell death appear to support the reviewers’ view in regards to a lack of extensive activation of cell death pathways in neurons, especially when compared to other cell types in WNV-infected CNS (i.e., infiltrating T and B cells). In addition, a CNS-region-specific activation of the programmed cell death in infiltrated lymphocytes would suggest the initiation of the resolution phase of the acute cellular inflammatory responses in WNV-infected CNS. This is consistent with our previously published results with NHP model of other flavivirus infections of the CNS (23), where we showed that the perivascular infiltrated lymphocytes rather than neurons were undergoing apoptosis.

Discussions of the above findings were added to the Discussion.

7) That the genes which are being regulated have multiple functions, i.e. are pleiotropic, does not seem particularly surprising. The authors argue this leads to unintended deleterious effects but the logical connection there is not clear, why could it not be the opposite? Actually providing a benefit?

If the concept of pleiotropy does not seem particularly surprising, the concept of an unintended detrimental effect of induction of the pleiotropic genes that regulate immune/defense responses to virus infection but also have distinct functions in neurons, their synapses, and neurotransmission, should be clear. We argue that WNV infection of the CNS disrupts normal homeostasis of the immune-neural-synaptic axis via induction of pleiotropic gene regulation of host responses with possible off-target effects of virus-induced host immune responses on neural functions and neurotransmission. We believe that our Discussion provides a sufficient logical explanation of these concepts.

“CNS neurons are terminally differentiated cells and cannot be replenished, thus virus infection of neurons present a challenge for both the nervous and immune system to coordinate virus clearance while protecting neurons and preserving neural function. […] Remarkably, some pleiotropic proteins, such as proinflammatory cytokines and proteins in the complement system and major histocompatibility complex are essential for the establishment, organization, function, and removal of synapses between neurons (24).”

“We propose a scenario in which changes in expression of pleiotropic genes that are intended to activate and maintain immune responses to virus infection would have unintended and potentially devastating off-target effects on neuron-to-neuron synapses and chemical neurotransmission”.

“Since WNV can spread transsynaptically (20), changes in synaptic homeostasis identified in this study may indicate either a direct damaging impact of infection on neurotransmission or an attempt by the host to arrest virus dissemination and compensate for changes in function. More studies will be required to understand to what extent these synaptic changes represent pathological, protective, and/or compensatory mechanisms, as well as whether they are reversible”.

“In summary, our findings add a new dimension to understanding of regulation of the immune-neural-synaptic axis and how its homeostasis is altered during virus infection in primates. […] Since activation of expression of the pleiotropic genes reported here may be a part of conserved host immune responses to many viral infections, our data may serve as a resource in the search for new therapeutic approaches to restore homeostasis in interactions between the nervous and immune system at the time when virus has been cleared from the CNS”.

References:

1) Maximova OA, et al. (2014) Assurance of neuroattenuation of a live vaccine against West Nile virus: A comprehensive study of neuropathogenesis after infection with chimeric WN/DEN4Delta30 vaccine in comparison to two parental viruses and a surrogate flavivirus reference vaccine. Vaccine 32(26):3187-3197.

2) Maximova OA, et al. (2008) Comparative neuropathogenesis and neurovirulence of attenuated flaviviruses in nonhuman primates. Journal of virology 82(11):5255-5268.

3) Messaoudi I, Estep R, Robinson B, and Wong SW (2011) Nonhuman primate models of human immunology. Antioxidants and redox signaling 14(2):261-273.

4) Lemon RN and Griffiths J (2005) Comparing the function of the corticospinal system in different species: organizational differences for motor specialization? Muscle and nerve 32(3):261-279.13

5) Perry VH and O'Connor V (2010) The role of microglia in synaptic stripping and synaptic degeneration: a revised perspective. ASN neuro 2(5):e00047.

6) Koopmans F, et al. (2019) SynGO: An Evidence-Based, Expert-Curated Knowledge Base for the Synapse. Neuron.

7) Vasek MJ, et al. (2016) A complement-microglial axis drives synapse loss during virus-induced memory impairment. Nature 534(7608):538-543.

8) Weinhard L, et al. (2018) Microglia remodel synapses by presynaptic trogocytosis and spine head filopodia induction. Nature communications 9(1):1228.

9) Di Liberto G, et al. (2018) Neurons under T Cell Attack Coordinate Phagocyte-Mediated Synaptic Stripping. Cell 175(2):458-471.e419.

10) Perry VH and O'Connor V (2008) C1q: the perfect complement for a synaptic feast? Nature reviews. Neuroscience 9(11):807-811.

11) Eroglu C and Barres BA (2010) Regulation of synaptic connectivity by glia. Nature 468(7321):223-231.

12) Verkhratsky A and Nedergaard M (2014) Astroglial cradle in the life of the synapse. Philosophical transactions of the Royal Society of London. Series B, Biological sciences 369(1654):20130595.

13) Chung WS, et al. (2013) Astrocytes mediate synapse elimination through MEGF10 and MERTK pathways. Nature 504(7480):394-400.

14) Troscher AR, et al. (2019) Microglial nodules provide the environment for pathogenic T cells in human encephalitis. Acta neuropathologica 137(4):619-635.

15) Kreutzfeldt M, et al. (2013) Neuroprotective intervention by interferon-gamma blockade prevents CD8^+^ T cell-mediated dendrite and synapse loss. The Journal of experimental medicine 210(10):2087-2103.

16) Stephan AH, Barres BA, and Stevens B (2012) The complement system: an unexpected role in synaptic pruning during development and disease. Annual review of neuroscience 35:369-389.

17) Garber C, et al. (2019) T cells promote microglia-mediated synaptic elimination and cognitive dysfunction during recovery from neuropathogenic flaviviruses. Nature neuroscience 22(8):1276-1288.

18) Pogodina VV, et al. (1983) Study on West Nile virus persistence in monkeys. Archives of virology 75(1-2):71-86.

19) Raudvere U, et al. (2019) g:Profiler: a web server for functional enrichment analysis and conversions of gene lists (2019 update). Nucleic acids research 47(W1):W191-w198.

20) Maximova OA, Bernbaum JG, and Pletnev AG (2016) West Nile Virus Spreads Transsynaptically within the Pathways of Motor Control: Anatomical and Ultrastructural Mapping of Neuronal Virus Infection in the Primate Central Nervous System. PLoS neglected tropical diseases 10(9):e0004980.

21) Cho H, et al. (2013) Differential innate immune response programs in neuronal subtypes determine susceptibility to infection in the brain by positive-stranded RNA viruses. Nature medicine 19(4):458-464.

22) Omalu BI, Shakir AA, Wang G, Lipkin WI, and Wiley CA (2003) Fatal fulminant pan-meningo-polioencephalitis due to West Nile virus. Brain pathology (Zurich, Switzerland) 13(4):465-472.

23) Maximova OA, Faucette LJ, Ward JM, Murphy BR, and Pletnev AG (2009) Cellular inflammatory response to flaviviruses in the central nervous system of a primate host. The journal of histochemistry and cytochemistry : official journal of the Histochemistry Society 57(10):973-989.

24) Boulanger LM (2009) Immune proteins in brain development and synaptic plasticity. Neuron 64(1):93-109.